# Disability and quality of life assessment using WHODAS-12 items 2.0 and EQ-5D-5L in a rural area endemic for loiasis in the Republic of Congo: A population-based cross-sectional study (the MorLo project)

Marlhand C. Hemilembolo[1,2], Jérémy T. Campillo[1], Valentin Dupasquier[3], Elodie Lebredonchel[4], Samuel Beneteau[1], Ludovic G. Rancé[5], Glorifié Madoulou Moulabou[2], Ange Clauvel Niama[6], Richard R. Bileckot[6], Sébastien D. S. Pion[1], Michel Boussinesq[1], François Missamou[2], Cédric B. Chesnais [1,2]*

1 Université de Montpellier, TransVIHMI, INSERM, Institut de Recherche pour le Développement (IRD), Montpellier, France, 2 Programme National de Lutte contre l'Onchocercose (PNLO), Direction de l'Épidémiologie et de la Lutte contre la Maladie, Ministère de la Santé et de la Population, Brazzaville, Republic of Congo, 3 Department of Cardiology, Montpellier University Hospital, Montpellier, France, 4 Department of Biochemistry, Hôpitaux Universitaires Paris Nord Val de Seine – site Bichat, Assistance Publique des Hôpitaux de Paris, Paris, France, 5 Department of Anesthesiology and Critical Care Medicine, Montpellier University Hospital, Montpellier, France, 6 Faculté des Sciences de la Santé, Université Marien Ngouabi, Brazzaville, Republic of Congo

* cedric.chesnais@ird.fr

## Abstract

### Background

*Loa loa* filariasis, a parasitic infection endemic to Central Africa, is a common cause of medical consultation in this region.

### Methods

To evaluate the quality of life (QoL) of individuals living in loiasis-endemic areas, we enrolled 991 subjects (one-third being microfilaremic) from the general population of a rural area in the Republic of Congo. QoL and disability were assessed using WHO-DAS 2.0 12-items and EQ-5D-5L questionnaires. We collected data on the number of eye worm (Ew) and Calabar swelling episodes experienced throughout their lives, as well as individual *L. loa* microfilarial densities and information on infections with soil-transmitted helminthiasis.

### Principal findings

Individuals with a history of Ew had a nearly doubled risk of experiencing at least moderate disability (score >25/100) (adjusted Odds-Ratio = 1.77 (95%CI [1.05-2.99], p = 0.033), compared to those without such a history. Those with more than 10

**Data availability statement:** The data that support the findings of this study are deposited with restricted access in DataSuds repository (IRD, France) at https://doi.org/10.23708/KCEDO2. They cannot be publicly shared because of legal restriction. The access to the data is subject to approval and a data sharing agreement. Related documentations are openly available and granted under CC-BY license.

**Funding:** This work was supported by the HORIZON EUROPE European Research Council (ERC) 2020 [grant agreement No 949963]. CBC is the carrier of this grant. The funders had no role in study design, data collection and analysis, decision to publish, or preparation of the manuscript.

**Competing interests:** The authors have declared that no competing interests exist.

episodes of Ew during their lifetime had a 28% increase in overall disability as measured by WHODAS. No other variable related to loiasis (Calabar swelling frequency, *L. loa* microfilarial density and positivity to *L. loa* antibody rapid test) was associated with the various scores. Additionally, infection with *Trichuris trichiura* was associated with worse anxiety score (adjusted incidence risk ratio = 1.22 (95% CI [1.06-1.39], p = 0.004)).

## Conclusions

The impact of loiasis on daily QoL appears to be primarily due to adult worms rather than microfilarial density. Indeed, our findings strongly suggest that the number of Ew episodes, likely reflecting the cumulative burden of adult worms, is the main correlate of worse QoL scores. These episodes seem to affect multiple dimensions of functioning, with notable impact on mobility, pain, anxiety, and daily activities. In contrast, microfilariae would primarily induce organ dysfunction. Further studies are needed to better understand the respective clinical impacts of adult worms and *L. loa* microfilariae.

## Author summary

Recent publications seem to suggest a real effect of loiasis on health. Until now, most studies have prioritized organ-by-organ evaluations, but in this article, we assessed the quality of life (QoL) of individuals living in loiasis-endemic areas. To do this, standardized tools (WHODAS 2.0 12-items and EQ-5D-5L) were applied. For the first time, we report QoL assessments in a population living in a rural area of the Republic of Congo. The average score indicated a moderate level of disability, mainly due to pain, anxiety disorders, cognitive issues, and mobility problems. We were able to show, for the first time, the possible impact of adult *Loa loa* worms on reduced QoL, primarily through mobility impairments. By contrast, we did not identify any effect of *L. loa* microfilaremia. Finally, and surprisingly, we identified a signal suggesting an impact of *Trichuris trichiura* on anxiety, which warrants further investigation to confirm or refute this finding.

## Introduction

Loiasis is a parasitic disease caused by *Loa loa*, a filarial nematode transmitted from human to human by tabanids (*Chrysops* spp.). *L. loa* is endemic only in Africa, from southeast Benin to South Sudan and from Chad to Angola [1]. Demographic data suggest that tens of millions people are exposed to the parasite and about 15 million are actually infected [2]. Loiasis is known to cause benign manifestations, such as pruritus, subconjunctival migration of the adult worms ("eye worm"), and transient episodes of angioedema ("Calabar swellings"), and is a frequent reason for medical

consultation [3,4]. Based on these symptoms alone, loiasis has been reported to have a considerable public health impact, with its estimated burden of disease in Gabon being close to that of schistosomiasis [5].

In addition, loiasis can induce complications affecting the cardiovascular system, the central nervous system, the spleen and the kidneys [6–9]. These results could partly explain the excess mortality observed in microfilaremic individuals (i.e., those who show microfilariae [mf], the larval progeny stages of the parasite, in their blood) in two retrospective cohort studies performed in Cameroon and the Republic of Congo [10,11]. Despite the high endemicity levels of loiasis in Central Africa and the fact that some manifestations might detrimentally affect everyday activity, no study has ever been conducted to evaluate the impact of this disease on quality of life (QoL). Since microfilariae appear to be associated with increased mortality and severe organ complications, it is possible that microfilarial density (MFD) is linked to an overall impaired QoL, potentially reflecting an advanced state of fatigue resulting from specific complications (e.g., renal failure). Furthermore, as no study to date has demonstrated an association between adult worms and severe complications, it is difficult to extrapolate their potential effect on QoL. However, a study conducted in Gabon found an association between loiasis (defined as either microfilaremia or a history of adult worms, based on eye worm or Calabar swelling) and general symptoms such as headaches, paresthesia, and arthralgia [4]. Adult worms, due to their migration, could contribute to symptoms related to chronic pain or mobility disorders.

Health and well-being assessments are crucial for understanding the impact of illnesses and health conditions on individuals' QoL. The World Health Organization Disability Assessment Schedule-12-items (WHODAS 2.0) score is a widely used tool to assess overall functioning and health-related disability across different populations and in different life conditions. In Africa, it has been used in patients with multiple morbidity from chronic non-communicable diseases in Ethiopia [12] and in stroke patients in South Africa [13]. It has also been applied to patients with neglected tropical diseases (NTDs), such as those suffering from Buruli ulcer in Ghana [14], podoconiosis in Ethiopia [15], onchocerciasis in Nigeria [16], or in the assessment of care for lymphatic filariasis [17,18]. Concurrently, the EuroQol Five Dimensions (EQ-5D) score provides a multidimensional measure of health that integrates both physical and psychological aspects of QoL. In Africa, this questionnaire has been used to measure health-related QoL in patients with type 2 diabetes in Burkina Faso [19] or in patients with tuberculosis in rural areas of Malawi [20], and a specific value set was developed in Uganda [21].

The aim of this study was to apply both the WHODAS 2.0 score and the EQ-5D-5 levels (EQ-5D-5L) score to assess the QoL in a population living in a loiasis endemic area of the Republic of Congo and to evaluate the impact of loiasis on the results.

## Materials and methods

### Ethics statement

This study received approval from the Ethics Committee of the Congolese Foundation for Medical Research (N° 036/CIE/FCRM/2022) and the Congolese Ministry of Health and Population (N° 376/MSP/CAB/UCPP-21). All participants received clear and appropriate information in Kikongo, Lingala, or in local vernacular language, and signed an informed consent form for this specific study.

### Study design

This cross-sectional study is part of the Morbidity due to Loiasis (MorLo) project, an international collaborative cohort study aimed at evaluating the prevalence and incidence of *L. loa*-related organ-specific complications in rural areas of Central Africa. It was conducted at the baseline of the cohort from May 16 to June 11, 2022 and involved individuals living in 21 villages located near Sibiti, the capital town of the Lékoumou division of the Republic of Congo. This region is endemic for loiasis (overall prevalence of 20% for microfilaremia [22]), and hypo-meso endemic for genito-urinary schistosomiasis [23].

The general design, the study area and the characteristics of the population involved in MorLo in the Republic of Congo have been described elsewhere [7,8]. Briefly, the inclusion criteria were: residency in the study area since 2019, age 18 or older, and prior screening for *L. loa* microfilaremia in 2019 as part of a participant selection survey for a previous clinical trial [24] (Fig 1). Individuals with more than 500 *L. loa* microfilariae/mL in 2019 were matched by sex and age (within a five-year range) with two amicrofilaremic individuals from the same village. However, a new thick blood smear (TBS) was made in 2022 at the time of this study, and the results of these were used for our analyses.

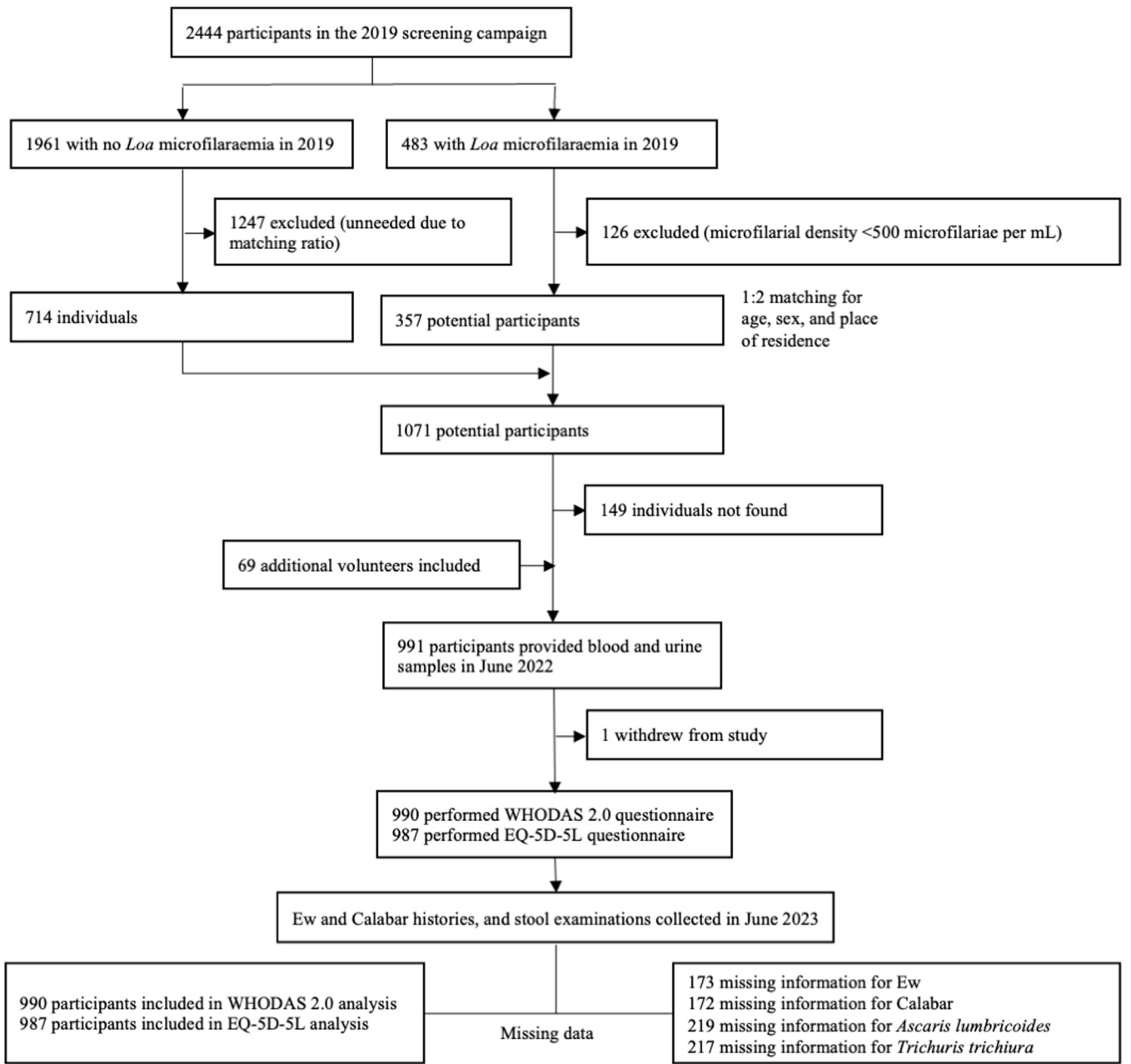

**Fig 1. Flowchart of inclusion procedure (adapted from [8]).**

## Laboratory procedures

Fifty microliters (µL) of blood were collected from each subject by finger-prick with a sterile lancet between 10 am and 4 pm to prepare a TBS. TBS were stained with Giemsa and examined under a microscope at 100×magnification by two experienced technicians to count the *L. loa* and *Mansonella perstans* mf (*M. perstans* is another filarial species endemic in the study area). Each TBS was independently read once by each technician and the arithmetic mean of the counts was used for the statistical analyses. In case of discrepancy between the results of the two readings (±20%), the slide was read a third time and the two closest results were averaged. Venous blood collected in a heparinized tube was used to perform a *L. loa* Antibody Rapid Test (*Loa* ART; Drugs & Diagnostics for Tropical Diseases, San Diego, CA) – the test line was compared to a 10-color scale to measure intensity [25].

Participant's past exposure to *Onchocerca volvulus* was assessed using an Ov16 Rapid Diagnostic Test (Biplex *L. loa*/Ov16 RDT; Drugs & Diagnostics for Tropical Diseases, San Diego, CA). Two small skin biopsies (skin snips) were collected using a 2 mm Holth-type corneoscleral punch from the iliac crests of each individual with positive Ov16 RDT and incubated in saline at room temperature for 24 hours. The incubation medium was examined under a microscope for the presence or absence of *O. volvulus* mf.

Sickle cell status (HbAA, HbAS or HbSS) was assessed using a rapid test (HemoTypeSC, HT401RUO-USA) performed on whole blood collected in EDTA tubes [26–28] and read after 10 minutes. Eosinophilia's measurements were performed from whole blood collected in EDTA tubes using the HemoCue WBCDiff device (HemoCue France, Serris, France). In addition, we performed a urinalysis and, in case of hematuria, a rapid diagnostic test (Schistosoma ICT IgG-IgM, LDBIO Diagnostics, Lyon, France) was performed and a filter paper through which urine was passed with a syringe was stained with Lugol's iodine and examined under a microscope to detect eggs of *Schistosoma haematobium*.

Finally, during the 1-year follow-up cohort assessment, we offered to the participants the option to provide stool samples for soil-transmitted helminthiasis (STH) and intestinal schistosomiasis screening. STH infections were identified by microscopic examination of stool specimens. Participants were supplied with a 50-mL plastic container and instructed to collect a morning stool sample. The collected containers were placed in cooling boxes and shipped to the laboratory within 6 hours. Upon arrival, the samples were either immediately processed or stored overnight at 6°C. Using the Kato-Katz method, a thick smear was prepared from each stool sample and these smears were examined under a microscope at 40×magnification.

## Demographic and health data collection

Age, sex, occupation, marital status, level of education (number of years of schooling), mean blood pressure, and body mass index (BMI) were recorded for each participant at baseline. In addition, information on the number of eye worm (Ew) episodes and the number of Calabar swelling episodes during their lifetime was recorded during the 1-year follow-up cohort assessment (the former was assumed to represent a proxy of the number of adult worms parasitizing the subject). Information on history of Ew and Calabar swelling was collected by surveyors experienced in working with the National Onchocerciasis Control Program. Regarding Ew, the standard RAPLOA procedure was used: participants were first asked whether they had experienced an Ew episode, and then whether the condition they had experienced was similar to the one shown on a color picture of an eye with an Ew [1].

## Disability and quality of life assessment

The questionnaires were presented and discussed with local authorities and community health workers. This step was essential to ensure that certain terms were understandable to the population. Once completed, the two questionnaire supervisors were trained to administer the questions in a standardized manner.

WHODAS 2.0 questionnaire (S1 Text) is a standardized tool for assessing health and disability across various conditions, populations, and cultures. All 12 items pertain to activities undertaken in the past month and are rated on a scale

of 0–4. We assessed both the total score (global score) and the results within the six domains of functioning: "cognition" (defined as understanding and communicating – items 3 plus 6 of the questionnaire), "mobility" (defined as ability to move and get around – items 1 plus 7), "self-care" (defined as related to hygiene, dressing, and eating – items 8 plus 9), "social" (defined as getting along with others – items 10 plus 11), "household" (defined as ability to attend to everyday responsibilities – items 2 plus 12), and "participation" (defined as participation in society – items 4 plus 5). Higher scores reflect greater disability, with the global score ranging from 0 to 48 (sum of the 12 items - no disability to complete disability) and individual domain scores ranging from 0 to 8 (sum of the two items defining the domain - no disability to complete disability).

Firstly, for the descriptive analysis, and to facilitate interpretation, we transformed the individual domain scores (0–8) and the global score (0–48) to get values from 0 (no disability) to 100 (full disability) [29]. Participants were then asked questions H1-H3 of the questionnaire to assess the extent to which the various difficulties they encountered over the last 30 days affected their lives. They were asked to provide the number of days the difficulties were present (H1, indicating mild disability), the number of days they were totally unable to carry out their usual activities or work because of any health condition (H2, indicating full disability) and the number of days (not counting the days they were totally unable) they cut back or reduced their usual activities or work because of any health condition (H3, indicating partial disability).

Health-related QoL was also assessed using the EQ-5D-5L score (S2 Text) with the "eq5d" R package [30]. The health state profile of each individual was first described by combining (in a 5-digit code) the response levels for each of the 5 items (called "dimensions") of the questionnaire: "mobility", "autonomy", "usual activities", "pain/discomfort", and "anxiety/depression" (in this order). For each dimension ("EQmobility", "EQautonomy", "EQdaily", "EQpain", "EQanxiety") response levels can range from 1 (no problem) to 5 points (unable to) and the five-digit number for a given individual can range from 11111 (full health) to 55555 (worst health). As for the WHODAS 2.0 questionnaire, for the descriptive analysis, and to facilitate interpretation, we transformed the individual dimensions scores to get values from 0 (no disability) to 100 (full disability). Since no EQ-5D-5L standard value set has been developed for the Republic of Congo, we chose to use the results obtained during the development of a "lite" version of the EuroQol Valuation Technology (EQ-VT) in Uganda [21] to calculate the utility index score of each individual (reflecting how good or bad his/her health state is). This score (called here EuroQol Valuation Technology score, "EQvts") can range from -1 (worst QoL) to +1 (best QoL). Last, we asked participants to give a number between 0 (the worst state of health they could imagine) and 100 (the best state of health) to rate their own health status on a visual analog scale ("EQvas").

Table 1 summarizes the scoring systems and transformations applied to both questionnaires, distinguishing between the descriptive analyses and the modeling approach.

## Statistical analyses

**Power calculation.** The study cohort was established by including all identified microfilaremic individuals (approximately 330) within a 50 km radius around Sibiti, and selecting two amicrofilaremic individuals for each microfilaremic participant, resulting in a total sample of about 990 individuals. This sample size allowed sufficient statistical power to address the primary objective of the cohort: to detect with a 80% power an increased incidence of infectious diseases among microfilaremic compared to amicrofilaremic individuals over a 3-year follow-up period. In the present cross-sectional study, the sample size was not predetermined to detect differences in QoL due to a lack of prior data for calculation. However, by employing a 1:2 matching design (1 microfilaremic matched with 2 amicrofilaremic individuals) and incorporating 991 participants, we anticipated a robust statistical power of 100% to detect a twofold higher risk (odds ratio, OR) of at least moderate disability (i.e., a global WHODAS score >25) in microfilaremics compared to amicrofilaremics; and a power of approximately 90% to identify a 1.5-fold higher risk.

**Explanatory variables.** Fifteen explanatory variables were included in the models: age (into 10-year age categories), sex (male vs. female), marital status (as a couple vs. single), years of schooling (continuous), main declared occupation

**Table 1. Overview of scales and transformations applied to WHODAS 2.0 and EQ-5D-5L scores for descriptive and multivariable analyses.**

| Instrument | Score, domains, and dimensions | Original scale | Rescaled for descriptive analysis | Scale used in models | Family model |
|---|---|---|---|---|---|
| WHODAS | Global score | 4–48 | To 0–100 | 0–100 | Count model |
| | Cognition | 0–8 | To 0–100 | 0–8 | Count model |
| | Mobility | 0–8 | To 0–100 | 0–8 | Count model |
| | Selfcare | 0–8 | To 0–100 | 0–8 | Count model |
| | Social | 0–8 | To 0–100 | 0–8 | Count model |
| | Household | 0–8 | To 0–100 | 0–8 | Count model |
| | Participation | 0–8 | To 0–100 | 0–8 | Count model |
| | Mild disability | 0–30 | 0-30 | 0-30 | Count model |
| | Partial disability | 0–30 | 0-30 | 0-30 | Count model |
| | Full disability | 0–30 | 0-30 | 0-30 | Count model |
| | Moderate disability | NA | Yes/No | Yes/No | Logistic model |
| | Severe disability | NA | Yes/No | Yes/No | Logistic model |
| EQ-5D-5L | EQmobility | 1–5 | To 0–100 | 0–4 | Count model |
| | EQautonomy | 1–5 | To 0–100 | 0–4 | Count model |
| | EQdaily | 1–5 | To 0–100 | 0–4 | Count model |
| | EQpain | 1–5 | To 0–100 | 0–4 | Count model |
| | EQanxiety | 1–5 | To 0–100 | 0–4 | Count model |
| | EQvts | −1 to +1 | −1 to +1 | −1 to +1 | Linear model |
| | EQvas | 0–100 | 0–100 | 0–100 | Count model |

All scores (except EQvts and EQvas) indicate worse disability with increasing values, thus reflecting the proportion of disability for each domain or dimension. EQvts: standardized global score using set value from Uganda [21] as reference (this score can range from -1 [worst QoL] to +1 [best QoL]). EQvas: Participants' self-rating of their health status on the EQ-5D-5L visual analogue scale (this score can range between 0 [worst state of health] and 100 [best state of health]).

(farmer vs. other), mean blood pressure and BMI (continuous), loiasis RDT result (intensity of test line: 0, 1–2, 3–4, 5–6, and 7–10), sickle cell disease status (HbAA, HbAS, HbSS, and invalid), number of self-declared Ew episodes (0, 1–5, 6–10, > 10), number of self-declared episodes of Calabar swelling (0, 1–5, 6–10, > 10), *L. loa* MFD (0, 1–7,999 mf/mL, 8,000–19,999 mf/mL, and ≥20,000 mf/mL), intensity of *Ascaris lumbricoides* infection (0, 1–1,000, and >1,000 eggs per gram [epg]), presence of *Trichuris trichiura* infection (negative vs. positive), and eosinophilia level (≤2 × 10⁹ cells/L vs. > 2 × 10⁹ cells/L). For this latter, to improve specificity for potential associations with organ-related dysfunction and consequently with overall QoL, we chose an eosinophil cutoff of $2 \times 10^9$ cells/L, based on prior literature and clinical considerations specific to parasitic infections. Missing data were considered as a separate category when their percentages were >5%. *M. perstans*, onchocerciasis, hookworm as well as schistosomiasis results were not included in the analyses due to low numbers of positive individuals (see Results section).

**Internal consistency of the questionnaires.** The internal consistencies of the WHODAS 2.0 and EQ-5D-5L questionnaires were checked using the "psych" package in R software [31]. Even if Cronbach's alpha is the most commonly used coefficient to check internal consistencies, we chose to use the omega coefficient because the latter was shown to perform better [32]. We estimated the omega coefficients with a method using a principal axis factor analysis with one latent variable and three bootstrap replicates.

**Descriptive analyses.** Descriptive results for categorical variables were presented using means, standard deviations (SD), medians and interquartile ranges (IQR) in each category. In light of the significant number of judgment criteria and explanatory variables, no statistical test was conducted for the bivariate descriptive part. However, Spearman's correlation coefficients (ρ) using Bonferroni's corrections were calculated for continuous variables.

**Disability assessment.** As the global WHODAS 2.0 score, the score (scaled between 0 and 8) for each of the six domains, and the response to the three questions (H1-H3) regarding the number of days of disability are count variables, they were modeled using a Poisson model or a Negative Binomial model depending on the result of the likelihood ratio test for over-dispersion. The correlations between each explanatory variable were evaluated using the Spearman's coefficient (for continuous variables) or the Cramer's V test (for categorical variables). Multicollinearity was assessed using the variance inflation factor (VIF). An initial saturated model was performed, followed by a manual backward selection procedure using the global Wald test to retain only variables with $p < 0.100$ in the final models and/or including variables with potential modality of interest ($p < 0.100$). Final predictions of the total score were made for statistically significant variables. Finally, a logistic model was applied to explain a global WHODAS 2.0 score >25 (at least moderate global disability – or significant disability) and >50 (severe global disability) [29,33].

The EQvts was modeled using a saturated linear regression, followed by manual backward selection based on the global Wald test to retain only variables with $p < 0.100$ in the final models and/or including variables with potential modality of interest ($p < 0.100$). EQmobility, EQautonomy, EQdaily, EQpain, EQanxiety and EQvas being count variables (scaled between 0 and 4), they were modeled using a Poisson model or a Negative Binomial model depending on the result of the likelihood ratio test for over-dispersion.

Due to the substantial number of scores evaluated, we synthesized the interpretation of the results. For all variables of the WHODAS 2.0 questionnaire as well as for the five dimensions of the EQ-5D-5L questionnaire, higher scores indicate greater disability. In other words, for these scores, variables with an Incidence Rate Ratio (IRR) >1 are associated with worse scores. Conversely, for the evaluation of EQvts and EQvas, higher scores indicate better health status. Therefore, for these two scores, variables with a negative coefficient (for EQvts) or an IRR < 1 (for EQvas) are associated with worse scores.

Finally, in all our models, we evaluated the potential effect of polyparasitism and with our main exposure variables using likelihood ratio tests for the following interaction terms: *A. lumbricoides* and *Trichuris trichiura*, *A. lumbricoides* and *L. loa* MFD, *T. trichiura* and *L. loa* MFD, *A. lumbricoides* and Ew, *A. lumbricoides* and Calabar edema, *T. trichiura* and Ew, *T. trichiura* and Calabar edema, and a three-way interaction term between *A. lumbricoides*, *T. trichiura*, and Ew variables. Similarly, interaction terms between parasites (*A. lumbricoides*, *T. trichiura*, *L. loa* MFD, and Ew variables) and eosinophilia were assessed.

## Results

### Description of the study population

A total of 990 individuals participated in the study (1 individual withdrew consent). All of them completed the EQ-5D-5L questionnaire while only 987 were present at the WHODAS 2.0 assessment. Median age was 52 years (IQR = 40–62). Males accounted for 62.5% (n = 619). Participants were more often living as couples (63.5%) and had in general a low level of education (21.9% with 0 years of schooling). Among them, 353 (35.6%) were microfilaremic for *L. loa*.

Collection of information on Ew and Calabar swellings history, as well as stool examinations, were conducted only during the 1-year follow-up of the cohort (Fig 1). Consequently, we had 219 absences or refusals (22%) for stool examination, 173 for Calabar swellings history assessment, and 174 for Ew history. Out of 771 individuals (77.8% of those present at baseline) who volunteered for stool examinations, 391 tested positive for helminths eggs (50.7%). Among these, none presented hookworm or *Schistosoma* spp. eggs, 332 (43.1%) showed *A. lumbricoides* eggs, and 207 (26.8%) *T. trichiura* eggs in their stool, with 146 (18.8%) individuals exhibiting a co-infection. Six of the individuals with hematuria were positive for the *Schistosoma* RDT but none was positive for *S. haematobium* eggs in urine. Only six individuals (0.6%) had *M. perstans* mf in their blood (range: 20–660 mf/mL). Twenty-two individuals (2.2%) tested positive for Ov16 RDT but none of them had *O. volvulus* mf in the skin snips.

Regarding the Ew/Calabar history group, individuals with missing values were significantly younger than respondents (median age: 45 vs. 53 years, p<0.001), were more likely to live alone rather than with a partner (46.5% vs. 34.4%, p=0.002), and were less likely to report agriculture as their primary occupation (67.8% vs. 80.9%, p<0.001). They did not differ in terms of sex ratio (p=0.772), *L. loa* MFD (p=0.765), or years of education (p=0.623). Regarding the STH group, individuals with missing values were significantly younger than respondents (median age: 48 vs. 53 years, p=0.012), were more likely to live alone rather than with a partner (45.9 vs. 33.9%, p=0.001), and were less likely to report agriculture as their primary occupation (68.2% vs. 81.6%, p<0.001). They did not differ in terms of sex ratio (p=0.660), *L. loa* MFD (p=0.765), or years of education (p=0.958).

### Internal consistency of the questionnaires

The results of the WHODAS 2.0 questionnaire showed a high internal consistency with a total omega coefficient equal to 0.9, a correlation of scores with a factor equal to 0.97, and a multiple R square of scores with a factor equal to 0.94. Only two items (S10: Dealing with people you do not know? and S11: Maintaining a friendship?) were not well represented, with a Schmid-Leiman Factor loadings lower than 0.2. Regarding the EQ-5D-5L questionnaires, the total omega coefficient was 0.82, the correlation of scores with a factor was 0.92 and the multiple R square of scores with factors was 0.85. All items were well represented, with a Schmid-Leiman Factor loadings greater than 0.2.

### Correlations, multicollinearity, and interaction terms

*Trichuris trichiura* and *A. lumbricoides* infections showed correlation, with Cramer's V values of 0.4075 (and 0.7594 when accounting for missing data). Similarly, Ew and Calabar variables exhibited correlation (Cramer's V values >0.5), primarily due to missing data. High multicollinearity (VIF>10) was observed among both sets of variables, driven by shared missing values. No statistically significant association was found between age and the number of eye worm episodes (Chi$^2$ p=0.283; Cramer's V value of 0.0848), suggesting that the number of episodes does not increase with age.

Despite controlling for variables individually, none except Ew remained significant. Consequently, our final models were not susceptible to statistically significant correlations or substantial multicollinearity, ensuring interpretable results. Finally, among all our tests evaluating interaction terms, we found a significant interaction only for EQanxiety model between *T. trichiura* infection and eosinophilia (p=0.019), as well as a three-way interaction between T. trichiura, eosinophilia, and *L. loa* MFD (p=0.004). However, no significant interaction was observed between *L. loa* MFD and eosinophilia or with *T. trichiura* alone. To facilitate interpretation of this result, we presented in the main analysis the model without interaction terms. As a complementary analysis, we reported the interaction between *T. trichiura* and eosinophilia alone, stratified by *L. loa* MFD status (negative or positive).

### WHODAS 2.0 results

**Descriptive results.** The participants' mean score (/100) was 25.3 (SD 19.1). Table 2 and S1 and S2 Tables present the results for the categorical variables. Regarding continuous variables, the number of years of education was significantly and inversely correlated with the total score, with the cognition, mobility, self-care, household and participation individual domain scores, and with the number of days with mild disability and full disability (ρ=-0.33, -0.31, -0.33, -0.12, -0.33, -0.26, -0.33, and -0.28; p<0.001, <0.001, <0.001, 0.021, <0.001, <0.001, <0.001, and <0.001; respectively) (S1 Fig). BMI was not associated with any variable of interest. Finally, mean blood pressure was significantly associated with the mobility score and with the number of days with full disability (ρ=0.12 and 0.11; p=0.008 and 0.058; respectively).

The scores recorded for selfcare and social (WHODAS 2.0) showed that these domains represented a very low proportion of disability, whereas the mobility, participation, cognitive, and household domains had a higher percent of disability on our population study (S1 and S2 Tables, and Fig 2). Fig 2 illustrates the effect of sex on these disability indicators.

**Table 2.** Descriptive results of the main scores for WHODAS 2.0 and EQ-5D-5L questionnaires.

| | | Global WHODAS score/100 | | | | | | EQvts | | | | | | EQvas/100 | |
|---|---|---|---|---|---|---|---|---|---|---|---|---|---|---|---|
| | | N | Mean | SD | Median | P25 | P75 | N | Mean | SD | Median | P25 | P75 | Mean | SD |
| Total* | | 987 | 25.3 | 19.1 | 20.8 | 8.3 | 43 .8 | 990 | 0.4 | 0.5 | 0.6 | -0.0 | 0.9 | 67.1 | 19.2 |
| Age (y.o.) | 18-28 | 86 | 15.0 | 15.7 | 8.3 | 2.1 | 25.0 | 86 | 0.7 | 0.3 | 0.8 | 0.5 | 0.9 | 74.5 | 17.1 |
| | 29-38 | 137 | 20.1 | 17.2 | 14.6 | 6.2 | 35.4 | 137 | 0.5 | 0.4 | 0.7 | 0.2 | 0.9 | 71.8 | 18.5 |
| | 39-48 | 191 | 22.3 | 17.9 | 16.7 | 6.2 | 41.7 | 191 | 0.5 | 0.5 | 0.7 | -0.0 | 0.9 | 71.8 | 19.4 |
| | 49-58 | 253 | 23.2 | 17.7 | 18.8 | 8.3 | 37.5 | 255 | 0.5 | 0.5 | 0.7 | 0.0 | 0.9 | 68.6 | 18.8 |
| | 59-68 | 202 | 30.9 | 19.2 | 31.2 | 12.5 | 47.9 | 204 | 0.3 | 0.4 | 0.4 | -0.1 | 0.7 | 61.5 | 16.8 |
| | >68 | 118 | 38.8 | 18.7 | 43.8 | 25.0 | 50.0 | 118 | 0.1 | 0.5 | -0.0 | -0.3 | 0.6 | 54.7 | 17.7 |
| Sex | Female | 370 | 31.8 | 18.9 | 31.2 | 14.6 | 50.0 | 372 | 0.3 | 0.5 | 0.3 | -0.1 | 0.7 | 62.2 | 18.1 |
| | Male | 617 | 21.4 | 18.1 | 14.6 | 6.2 | 37.5 | 619 | 0.5 | 0.5 | 0.7 | 0.1 | 0.9 | 70.0 | 19.2 |
| Eye worm episodes | 0 | 423 | 23.8 | 19.4 | 18.8 | 6.2 | 41.7 | 425 | 0.5 | 0.5 | 0.7 | 0.0 | 0.9 | 68.3 | 18.9 |
| | 1-5 | 172 | 24.5 | 18.7 | 18.8 | 8.3 | 42.7 | 172 | 0.4 | 0.5 | 0.6 | -0.0 | 0.9 | 67.6 | 19.1 |
| | 6-10 | 140 | 27.2 | 17.3 | 27.1 | 12.5 | 42.7 | 141 | 0.4 | 0.4 | 0.5 | -0.0 | 0.7 | 64.3 | 18.0 |
| | >10 | 79 | 29.1 | 17.9 | 29.2 | 10.4 | 45.8 | 79 | 0.3 | 0.5 | 0.5 | -0.1 | 0.8 | 63.9 | 20.1 |
| | AMD | 173 | 26.6 | 20.2 | 22.9 | 8.3 | 45.8 | 174 | 0.4 | 0.5 | 0.5 | -0.0 | 0.8 | 67.1 | 20.2 |
| Calabar episodes | 0 | 568 | 24.6 | 19.3 | 20.8 | 8.3 | 41.7 | 570 | 0.4 | 0.5 | 0.6 | -0.0 | 0.9 | 67.2 | 19.0 |
| | 1-5 | 112 | 26.4 | 17.6 | 20.8 | 12.5 | 43.8 | 112 | 0.5 | 0.4 | 0.7 | 0.0 | 0.8 | 66.8 | 18.8 |
| | 6-10 | 88 | 26.1 | 18.5 | 26.0 | 8.3 | 43.8 | 89 | 0.4 | 0.5 | 0.4 | -0.0 | 0.9 | 67.2 | 19.3 |
| | >10 | 47 | 25.3 | 15.2 | 25.0 | 14.6 | 37.5 | 47 | 0.4 | 0.5 | 0.4 | -0.2 | 0.8 | 65.3 | 19.3 |
| | AMD* | 172 | 26.6 | 20.4 | 22.9 | 8.3 | 46.9 | 173 | 0.4 | 0.5 | 0.5 | -0.0 | 0.8 | 67.3 | 19.9 |
| *Loa* MFD (mf/mL) | 0 | 638 | 25.5 | 19.3 | 22.9 | 8.3 | 43.8 | 638 | 0.4 | 0.5 | 0.6 | -0.0 | 0.9 | 67.3 | 19.0 |
| | 1-7,999 | 252 | 25.5 | 18.9 | 20.8 | 8.3 | 43.8 | 256 | 0.4 | 0.5 | 0.6 | -0.0 | 0.9 | 65.7 | 19.7 |
| | 8,000-19,999 | 65 | 23.1 | 18.5 | 16.7 | 8.3 | 41.7 | 65 | 0.4 | 0.5 | 0.6 | -0.0 | 0.9 | 69.5 | 19.0 |
| | >19,999 | 32 | 24.5 | 15.8 | 28.1 | 9.4 | 35.4 | 32 | 0.4 | 0.4 | 0.5 | 0.0 | 0.8 | 67.2 | 18.2 |
| *Loa* RDT (Intensity)* | 0 | 59 | 24.6 | 18.2 | 18.8 | 8.3 | 39.6 | 59 | 0.4 | 0.5 | 0.5 | -0.1 | 0.8 | 64.3 | 19.9 |
| | 1-2 | 102 | 24.7 | 19.7 | 18.8 | 6.2 | 45.8 | 102 | 0.4 | 0.5 | 0.6 | -0.0 | 0.9 | 68.2 | 19.7 |
| | 3-4 | 410 | 26.9 | 19.2 | 25.0 | 10.4 | 43.8 | 410 | 0.4 | 0.5 | 0.6 | -0.0 | 0.9 | 66.3 | 19.5 |
| | 5-6 | 354 | 23.6 | 18.5 | 18.8 | 8.3 | 39.6 | 358 | 0.5 | 0.5 | 0.6 | 0.1 | 0.9 | 68.4 | 18.7 |
| | >6 | 44 | 26.7 | 21.0 | 24.0 | 7.3 | 46.9 | 44 | 0.4 | 0.5 | 0.4 | -0.1 | 0.9 | 66.0 | 20.2 |
| Eosinophilia (× $10^9$ cells/L) | ≤ 2 | 766 | 24.7 | 18.9 | 20.8 | 8.3 | 41.7 | 769 | 0.4 | 0.5 | 0.6 | -0.0 | 0.9 | 67.5 | 19.5 |
| | > 2 | 148 | 25.4 | 19.4 | 20.8 | 8.3 | 42.7 | 149 | 0.5 | 0.5 | 0.7 | -0.0 | 0.9 | 67.4 | 18.7 |
| | AMD* | 73 | 31.7 | 19.6 | 35.4 | 12.5 | 50.0 | 73 | 0.3 | 0.5 | 0.3 | -0.1 | 0.8 | 61.9 | 15.8 |
| Main occupation | Other | 211 | 25.0 | 21.7 | 16.7 | 6.2 | 43.8 | 212 | 0.4 | 0.5 | 0.7 | -0.1 | 0.9 | 66.8 | 21.0 |
| | Farmer | 776 | 25.4 | 18.3 | 22.9 | 8.3 | 43.8 | 779 | 0.4 | 0.5 | 0.5 | -0.0 | 0.9 | 67.1 | 18.7 |
| Marital status | As a couple | 627 | 23.1 | 18.2 | 18.8 | 8.3 | 39.6 | 629 | 0.5 | 0.4 | 0.7 | 0.1 | 0.9 | 69.9 | 18.5 |
| | Single | 360 | 29.2 | 19.9 | 29.2 | 10.4 | 47.9 | 362 | 0.3 | 0.5 | 0.4 | -0.1 | 0.8 | 62.1 | 19.3 |
| Sickle cell disease status* | HbAA | 757 | 25.6 | 19.1 | 22.9 | 8.3 | 43.8 | 760 | 0.4 | 0.5 | 0.6 | -0.0 | 0.9 | 66.8 | 19.2 |
| | HbAS | 219 | 24.2 | 18.7 | 20.8 | 8.3 | 41.7 | 220 | 0.5 | 0.5 | 0.7 | 0.1 | 0.9 | 68.0 | 19.2 |
| Tobacco use* | No | 797 | 26.0 | 19.3 | 22.9 | 8.3 | 43.8 | 801 | 0.4 | 0.5 | 0.5 | -0.0 | 0.9 | 66.4 | 19.1 |
| | Yes | 182 | 22.8 | 18.2 | 16.7 | 6.2 | 39.6 | 182 | 0.5 | 0.5 | 0.7 | 0.0 | 0.9 | 69.5 | 19.3 |
| *Ascaris lumbricoides* (epg) | 0 | 437 | 24.0 | 18.1 | 20.8 | 8.3 | 39.6 | 439 | 0.5 | 0.5 | 0.6 | 0.1 | 0.9 | 68.6 | 18.3 |
| | 1-1,000 | 210 | 24.6 | 19.1 | 20.8 | 8.3 | 41.7 | 211 | 0.4 | 0.5 | 0.6 | -0.0 | 0.9 | 66.6 | 20.3 |
| | >1,000 | 121 | 28.4 | 20.1 | 27.1 | 10.4 | 43.8 | 121 | 0.3 | 0.5 | 0.4 | -0.1 | 0.8 | 63.6 | 18.4 |
| | AMD* | 219 | 27.0 | 20.2 | 22.9 | 8.3 | 45.8 | 220 | 0.4 | 0.5 | 0.5 | -0.0 | 0.8 | 66.4 | 19.9 |

*(Continued)*

**Table 2.** (Continued)

| | | Global WHODAS score/100 | | | | | | EQvts | | | | | | EQvas/100 | |
|---|---|---|---|---|---|---|---|---|---|---|---|---|---|---|---|
| | | N | Mean | SD | Median | P25 | P75 | N | Mean | SD | Median | P25 | P75 | Mean | SD |
| *Trichuris trichiura* infection | No | 564 | 24.4 | 18.6 | 20.8 | 8.3 | 41.7 | 566 | 0.4 | 0.5 | 0.6 | -0.0 | 0.9 | 68.2 | 19.0 |
| | Yes | 206 | 26.0 | 19.0 | 22.9 | 8.3 | 43.8 | 207 | 0.4 | 0.5 | 0.5 | -0.0 | 0.9 | 64.8 | 18.5 |
| | AMD* | 217 | 27.0 | 20.2 | 22.9 | 8.3 | 45.8 | 217 | 0.4 | 0.5 | 0.5 | -0.0 | 0.8 | 66.3 | 20.0 |

*AMD: absent/missing data. Other variables: total (4 missing data), RDT *loa* (18), Tobacco (8), Sickle cell disease (7 invalid results and 4 missing data). EQvts: standardized global score using set value from Uganda [21] as reference (this score can range from -1 [worst QoL] to +1 [best QoL]). EQvas: Participants' self-rating of their health status on the EQ-5D-5L visual analogue scale (this score can range between 0 [worst state of health] and 100 [best state of health]). SD: standard deviation. P25 and P75: 25th and 75th percentiles. MFD: microfilarial density. mf/mL: microfilariae per milliliter of blood. RDT: Rapid diagnostic test. epg: eggs per gram of stool.

**Multivariable analyses.** Across all models, older age was significantly associated with higher (worse) scores and higher numbers of days with disability (Table 3 and S3 Table for saturated models). Being a male was significantly associated with better scores, except for the number of days with partial and full disability. Consistently, longer education was positively associated with better outcomes for all indicators. Single individuals demonstrated worse results for all scores except for mobility, selfcare, and social items. Similarly, in cases of full and partial disability, the number of days impacted during the month was significantly higher for single individuals than for individuals living as a couple. Finally, no significant effects were noted for STH, tobacco use or for individuals with an HbAS profile.

Although the average number of partial disability days increased with the frequency of Calabar swelling (S2 Table), neither a history of Calabar swellings nor *L. loa* RDT results were associated with any WHODAS 2.0 score in our multi-variable models. Individuals who had reported more than 10 episodes of Ew during their lifetime had significantly higher overall disability (adjusted incidence risk ratio aIRR = 1.28, 95%CI [1.04, 1.58], p = 0.023), and higher social disability (aIRR = 1.75, 95%CI [1.13, 2.70], p = 0.012) than those who said they had never experienced this sign. Additionally, we observed a gradient effect between the number of Ew episodes and mobility score: aIRR = 1.20, 95%CI [1.03, 1.41], p = 0.021, and 1.33, 95%CI [1.10, 1.62], p = 0.004, respectively, for individuals with 6–10 and more than 10 reported episodes during their lifetime, compared to those who reported no history of Ew episode. Although not significant (p < 0.100), individuals with more than 10 episodes of Ew during their lifetime had an increased aIRR for the household, cognitive and participation domains and for the number of days with mild disability. Note that the Ew variable exhibited a potential gradient effect for mild disability. Although it was not retained in our models, we observed a trend suggesting a relationship between the frequency of Ew and the full and partial disability variables (S3 Table). Fig 3 summarizes significant associations for WHODAS 2.0 questionnaire.

Based on this final model, predicted mean disability scores were 23.6 (95%CI 21.6–25.6) for individuals with no history of Ew, 24.3 (95%CI 21.1–27.6) for those with 1–5 episodes, 26.1 (95%CI 22.2–29.9) for those with 6–10 episodes, and 30.2 (95%CI 24.3–36.2) for those with more than 10 Ew episodes during their lifetime. This indicates a 28.0% deterioration in the overall score for individuals with the most Ew episodes.

Only sex, age, marital status, years of schooling, and Ew history were associated with at least moderate global disability (global WHODAS 2.0 score >25) and adjusted Odds-Ratio (aOR) values were 1.44 (95%CI [0.95-2.19], p = 0.085) and 1.77 (95%CI [1.05-2.99], p = 0.033), respectively for individuals with 6–10 episodes and those with more than 10 Ew episodes during their lifetime, compared to individuals with no history of Ew. Regarding the model for severe global disability (global WHODAS score >50), only sex, age, marital status, and years of schooling were statistically significantly and, therefore, retained in the final model.

All final models are illustrated in S2 Fig.

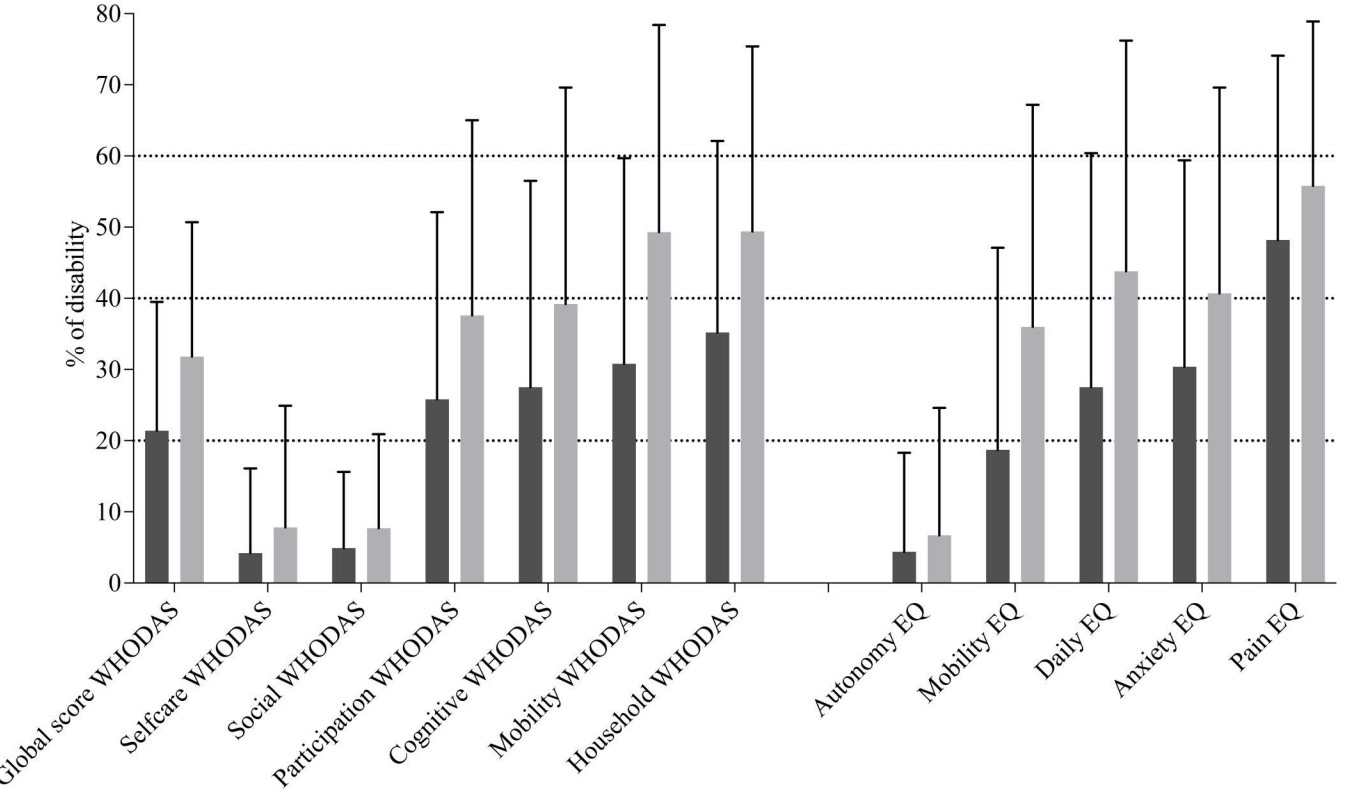

### EQ-5D-5L results

**Descriptive results.** Table 2 and S4 Table present the results for each categorical variable. EQvas assessment reported an average level of perceived good QoL of 67.1/100. Regarding continuous variables, the number of years of schooling was significantly and inversely correlated with the EQvts, EQmobility, EQautonomy, EQdaily, EQpain, EQanxiety, and EQvas (ρ = 0.31, -0.29, -0.12, -0.30, -0.24, -0.24, and 0.27; p < 0.001, < 0.001, 0.010, < 0.001, < 0.001, < 0.001, and < 0.001; respectively); BMI was not associated with any variable of interest; mean blood pressure tended to be significantly associated with EQmobility (ρ = 0.102, p = 0.067).

The proportion of disability was markedly lower for the EQautonomy score, and then gradually increased with the EQmobility, EQdaily, EQanxiety, and EQpain scores (Fig 2).

**Multivariable analyses.** Table 4 presents the results of the final models for each EQ 5D-5L dimension (S5 and S6 Tables for saturated models). Older age as well as being single were significantly associated with worse scores. Being male was significantly associated with better scores, except for EQanxiety, EQpain, and EQautonomy. Consistently, more years of schooling were positively associated with better outcomes across all indicators. Farmers exhibited better scores for EQvas but no significant association was found between this occupation and the results of the various EQ-5D-5L dimensions. *T. trichiura* infection was significantly associated with worse global EQvas (aIRR = 0.95, 95%CI [0.91, 0.99], p = 0.013) and EQanxiety scores (aIRR = 1.22, 95%CI [1.06, 1.39]). Regarding the significant interaction between eosinophilia and *T. trichiura* infection on EQanxiety scores, among individuals with positive *L. loa* microfilaremia,

**Fig 2. Disability levels by sex according to the WHODAS 2.0 domains and EQ-5D-5L dimensions.** Bars and error bars represent, respectively, the means and standard deviations. Black bars represent males, and grey bars represent females. Percentages (Y axis) indicate the average level of disability, first for the global WHODAS score (based on all items), and then individually for each domain or dimension.

**Table 3. Final multivariable analyses on WHODAS 2.0 score, the six domains, and the number of days with disabilities during the last month.**

| | | Global WHODAS 2.0 score | | Mobility | | Selfcare | |
|---|---|---|---|---|---|---|---|
| | | aIRR [95%CI] | p | aIRR [95%CI] | p | aIRR [95%CI] | p |
| Sex (Ref. female) | Male | 0.82 [0.72, 0.93] | 0.001 | 0.75 [0.67, 0.84] | <0.001 | 0.64 [0.43, 0.96] | 0.031 |
| Age (Ref. 18–28 y.o.) | 29-38 | 1.29 [1.01, 1.64] | 0.044 | 1.51 [1.16, 1.96] | 0.002 | 1.70 [0.67, 4.29] | 0.262 |
| | 39-48 | 1.36 [1.08, 1.72] | 0.009 | 1.58 [1.23, 2.02] | <0.001 | 2.26 [0.94, 5.43] | 0.069 |
| | 49-58 | 1.49 [1.19, 1.87] | <0.001 | 1.69 [1.33, 2.15] | <0.001 | 1.95 [0.83, 4.58] | 0.124 |
| | 59-68 | 1.66 [1.31, 2.11] | <0.001 | 1.98 [1.55, 2.54] | <0.001 | 2.88 [1.20, 6.92] | 0.018 |
| | >68 | 1.96 [1.50, 2.54] | <0.001 | 2.42 [1.86, 3.15] | <0.001 | 4.09 [1.63, 10.31] | 0.003 |
| Eye worm episodes (Ref. 0)[a] | 1-5 | 1.03 [0.88, 1.21] | 0.694 | 1.07 [0.92, 1.24] | 0.395 | | |
| | 6-10 | 1.10 [0.93, 1.31] | 0.253 | 1.20 [1.03, 1.41] | 0.021 | | |
| | >10 | 1.28 [1.04, 1.58] | 0.023 | 1.33 [1.10, 1.62] | 0.004 | | |
| | AMD* | 1.17 [1.00, 1.38] | 0.049 | 1.20 [1.03, 1.39] | 0.018 | | |
| Marital status (Ref. as a couple) | Single | 1.12 [0.99, 1.26] | 0.061 | | | | |
| Years of schooling (continuous) | | 0.96 [0.95, 0.98] | <0.001 | 0.97 [0.95, 0.98] | <0.001 | | |

| | | Household | | Cognitive | | | |
|---|---|---|---|---|---|---|---|
| | | aIRR [95%CI] | p | aIRR [95%CI] | p | | |
| Sex (Ref. female) | Male | 0.86 [0.78, 0.94] | 0.001 | | | | |
| Age (Ref. 18–28 y.o.) | 29-38 | 1.21 [0.99, 1.48] | 0.068 | 1.28 [0.92, 1.77] | 0.137 | | |
| | 39-48 | 1.35 [1.11, 1.63] | 0.002 | 1.44 [1.06, 1.96] | 0.021 | | |
| | 49-58 | 1.51 [1.25, 1.82] | <0.001 | 1.60 [1.19, 2.15] | 0.002 | | |
| | 59-68 | 1.59 [1.31, 1.92] | <0.001 | 1.79 [1.31, 2.44] | <0.001 | | |
| | >68 | 1.79 [1.46, 2.19] | <0.001 | 2.07 [1.48, 2.89] | <0.001 | | |
| Eye worm episodes (Ref. 0)[a] | 1-5 | 0.98 [0.88, 1.10] | 0.780 | 1.06 [0.87, 1.29] | 0.566 | | |
| | 6-10 | 1.03 [0.91, 1.17] | 0.620 | 1.10 [0.89, 1.36] | 0.369 | | |
| | >10 | 1.14 [0.98, 1.33] | 0.090 | 1.24 [0.96, 1.62] | 0.100 | | |
| | AMD* | 1.11 [0.99, 1.25] | 0.074 | 1.22 [1.00, 1.49] | 0.045 | | |
| Marital status (Ref. as a couple) | Single | 1.09 [1.00, 1.19] | 0.060 | 1.21 [1.04, 1.39] | 0.011 | | |
| Years of schooling (continuous) | | 0.96 [0.95, 0.98] | <0.001 | 0.95 [0.93, 0.96] | <0.001 | | |

| | | Social | | Participation | | Mild disability | |
|---|---|---|---|---|---|---|---|
| | | aIRR [95%CI] | p | aIRR [95%CI] | p | aIRR [95%CI] | p |
| Sex (Ref. female) | Male | 0.57 [0.44, 0.74] | <0.001 | 0.84 [0.74, 0.96] | 0.010 | 0.88 [0.77, 1.00] | 0.048 |
| Age (Ref. 18–28 y.o.) | 29-38 | 1.65 [0.96, 2.84] | 0.072 | 1.14 [0.86, 1.50] | 0.370 | 1.24 [0.95, 1.60] | 0.113 |
| | 39-48 | 1.08 [0.63, 1.85] | 0.780 | 1.21 [0.93, 1.58] | 0.153 | 1.38 [1.08, 1.77] | 0.011 |
| | 49-58 | 0.95 [0.56, 1.60] | 0.842 | 1.29 [1.00, 1.66] | 0.051 | 1.58 [1.24, 2.01] | <0.001 |
| | 59-68 | 1.38 [0.81, 2.34] | 0.232 | 1.41 [1.08, 1.84] | 0.011 | 1.83 [1.42, 2.36] | <0.001 |
| | >68 | 1.18 [0.67, 2.07] | 0.567 | 1.76 [1.33, 2.34] | <0.001 | 2.02 [1.53, 2.66] | <0.001 |
| Eye worm episodes (Ref. 0) | 1-5 | 0.81 [0.56, 1.17] | 0.259 | 1.02 [0.86, 1.21] | 0.827 | 1.05 [0.89, 1.25] | 0.533 |
| | 6-10 | 0.83 [0.56, 1.23] | 0.346 | 1.12 [0.94, 1.34] | 0.213 | 1.12 [0.94, 1.35] | 0.208 |
| | >10 | 1.75 [1.13, 2.70] | 0.012 | 1.21 [0.97, 1.51] | 0.091 | 1.23 [0.98, 1.54] | 0.074 |
| | AMD* | 1.05 [0.74, 1.49] | 0.795 | 1.11 [0.93, 1.31] | 0.237 | 1.11 [0.94, 1.32] | 0.225 |
| Main occupation (Ref. Other) | Farmer | 0.68 [0.50, 0.92] | 0.014 | | | | |
| *Loa* MFD (mf/mL) (Ref. 0)[b] | 1-7,999 | 1.01 [0.76, 1.35] | 0.921 | | | | |
| | 8,000-19,999 | 1.09 [0.66, 1.82] | 0.728 | | | | |
| | >19,999 | 1.76 [0.93, 3.34] | 0.084 | | | | |
| Marital status (Ref. as a couple) | Single | | | 1.13 [0.99, 1.28] | 0.061 | | |
| Years of schooling (continuous) | | | | 0.96 [0.95, 0.98] | 0.000 | 0.96 [0.95, 0.98] | 0.000 |

| | | Full disability | | Partial disabililty | | | |
|---|---|---|---|---|---|---|---|

*(Continued)*

| | | Global WHODAS 2.0 score | | Mobility | | Selfcare | |
|---|---|---|---|---|---|---|---|
| | | aIRR [95%CI] | p | aIRR [95%CI] | p | aIRR [95%CI] | p |
| | | aIRR [95%CI] | p | aIRR [95%CI] | p | | |
| Sex (Ref. female) | Male | | | | | | |
| Age (Ref. 18–28 y.o.) | 29-38 | 1.50 [0.90, 2.47] | 0.117 | | | | |
| | 39-48 | 1.85 [1.15, 2.97] | 0.011 | | | | |
| | 49-58 | 2.32 [1.47, 3.66] | <0.001 | | | | |
| | 59-68 | 2.38 [1.47, 3.85] | <0.001 | | | | |
| | >68 | 3.18 [1.87, 5.43] | <0.001 | | | | |
| Marital status (Ref. as a couple) | Single | 1.46 [1.15, 1.85] | 0.002 | 1.61 [1.04, 2.49] | 0.032 | | |
| Years of schooling (continuous) | | 0.94 [0.91, 0.97] | <0.001 | | | | |

\* AMD: absent/missing data. aIRR: adjusted incidence risk ratio. 95% CI: 95% Confidence interval. MFD: microfilarial density. mf/mL: microfilariae per milliliter of blood. Mild disability (H1 question in the questionnaire); full disability (H2 question); Partial disability (H3 question). [a] Global Wald test for eye worm episodes variable: (global WHODAS 2.0 score, p=0.092; mobility, p=0.009; household, p=0.189; cognitive, p=0.226; social, p=0.030; participation, p=0.357; TI, p=0.341). [b] Global Wald test for *Loa* MFD, p=0.084. For all variables of the WHODAS 2.0 questionnaire, higher scores indicate greater disability. In other words, variables with an aIRR > 1 were associated with worse scores.

eosinophilia alone was associated with a decreased EQanxiety score (aIRR=0.57; 95%CI [0.38, 0.84], p=0.005), while *T. trichiura* infection alone showed a borderline association with increased EQanxiety (aIRR=1.24, 95%CI [0.98, 1.58], p=0.070). However, the combined presence of both eosinophilia and *T. trichiura* infection led to a significant worse EQanxiety scores (interaction term aIRR=2.01, 95%CI [1.11, 3.62], p=0.021). In contrast, among individuals without *L. loa* microfilaremia, there was no statistically significant interaction between eosinophilia and *T. trichiura* infection. Neither eosinophilia (aIRR=0.82, 95%CI [0.61, 1.10], p=0.193), *T. trichiura* infection (aIRR=1.10, 95%CI [0.90, 1.34], p=0.361), nor their interaction (aIRR=1.33, 95%CI [0.79, 2.22], p=0.281) were significantly associated with EQanxiety scores. However, a similar trend toward increased EQanxiety in co-infected individuals was observed, although it did not reach statistical significance in this subgroup. Last, no significant effects were noted for *A. lumbricoides* infection intensity and tobacco use.

The history of Calabar swellings, the *L. loa* MFD, and a positive *L. loa* RDT result were not significantly associated with any score. In contrast, individuals with more than 10 episodes of Ew during their lifetime tended to have a higher overall disability as measured by EQvas (aIRR=0.94, 95%CI [0.88, 1.00], p=0.061) and a worse anxiety score (aIRR=1.21, 95%CI [0.99, 1.49], p=0.062), compared to those without any history of Ew episodes. Notably, these individuals also demonstrated a significantly worse mobility score, with an average disability increase of 53% (aIRR=1.53, 95%CI [1.17, 2.00], p=0.007).

Fig 4 summarizes significant associations for EQ-5D-5L questionnaire and all final models are illustrated in S3 Fig.

## Discussion

This study presents, for the first time, QoL estimates for a rural population in the Republic of Congo. The combined use of the WHODAS 2.0 and the EQ-5D-5L enables a comprehensive assessment of their QoL, allowing for the consideration of specific disabilities represented by domains and dimensions, respectively. Overall, we observed that the global QoL of this study population is slightly impaired, with an average disability score of 25.3 (on a scale of 100) according to the WHODAS 2.0 results. Conversely, the average self-rated EQvas score in our study population was 32.9 (100-67.1), i.e., slightly higher than the WHODAS global score.

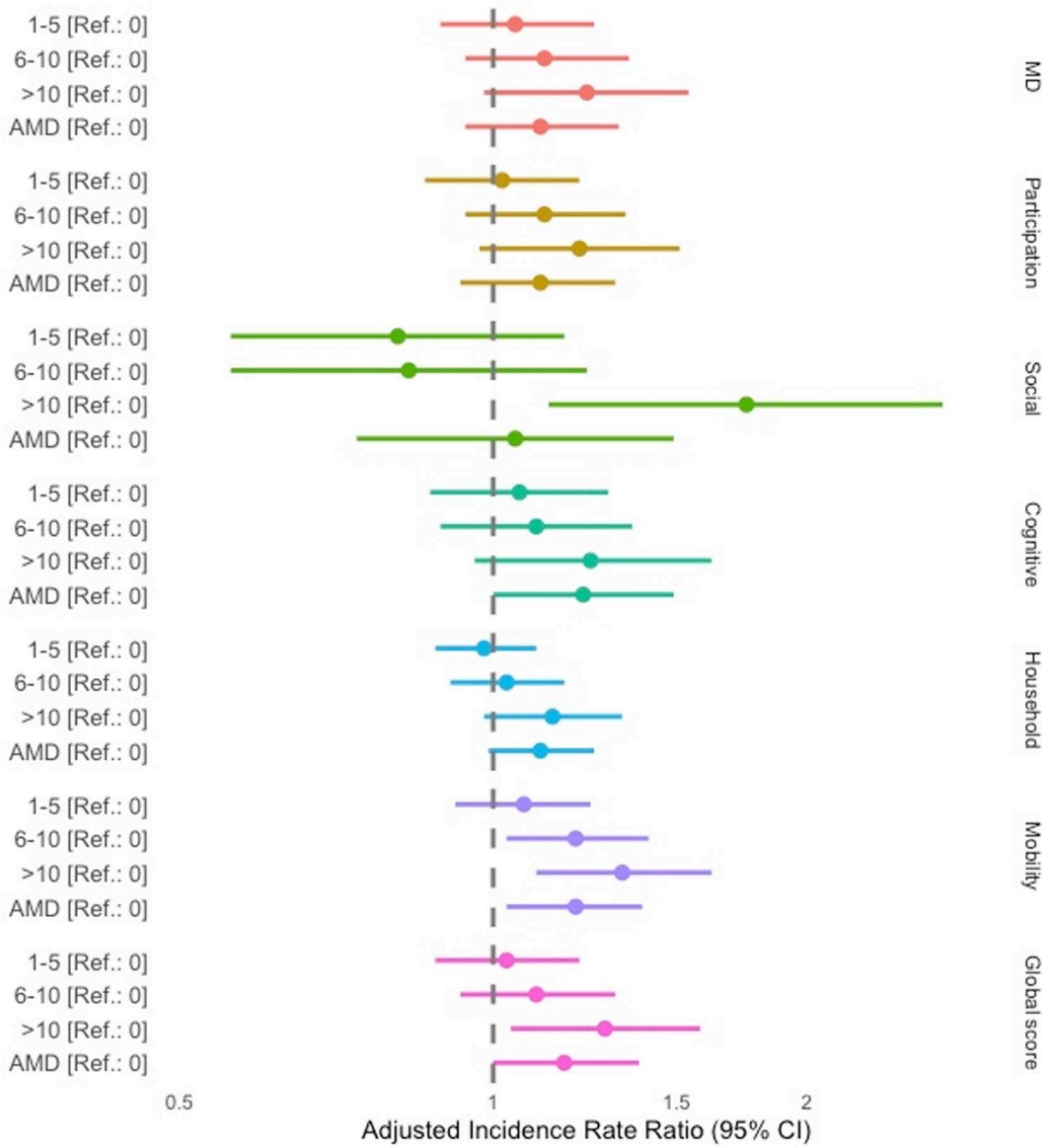

**Fig 3. Incidence Rate Ratios of eye worm episodes for all the WHODAS 2.0 indicators.** AMD: Absent/missing data; MD: mild disability. The horizontal bars represent 95% confidence intervals, and the dots indicate incidence rate ratios for each category compared to the reference (the reference value for eye worm episodes being 0). These results correspond to Table 2, which shows the final models where the Ew variable was retained (p<0.100).

We found well-identified variables (age, sex and number of years of schooling) that are commonly associated with QoL levels. Age emerged as a consistent predictor of deteriorated scores across multiple domains and higher number of days with disability. This aligns with existing literature highlighting the progressive nature of disability with advancing age due to factors such as physiological changes, accumulation of health conditions, and declining functional abilities. In our models, the association with age followed a clear gradient, with older participants—especially those over 68—showing significantly higher disability levels, particularly in mobility, cognition, and daily activities (WHODAS), as well as in mobility, pain/discomfort, and autonomy dimensions (EQ-5D-5L). These patterns illustrate the multifaceted impact of ageing on functioning

**Table 4. Final multivariable analyses on EQ-5D-5L scores.**

| | | EQvas[a] | | EQvts[a] | | EQmobility | | EQanxiety | |
|---|---|---|---|---|---|---|---|---|---|
| | | aIRR [95%CI] | p | ß-coef [95%CI] | p | aIRR [95%CI] | p | aIRR [95%CI] | p |
| Sex (Ref. female) | Male | 1.05 [1.01, 1.09] | 0.013 | 0.07 [0.01, 0.13] | 0.026 | 0.67 [0.57, 0.79] | 0.000 | | |
| Age (Ref. 18–28 y.o.) | 29-38 | 0.96 [0.89, 1.03] | 0.265 | −0.14 [−0.25, −0.02] | 0.025 | 2.02 [1.27, 3.22] | 0.003 | 1.24 [0.94, 1.62] | 0.122 |
| | 39-48 | 0.96 [0.89, 1.03] | 0.206 | −0.17 [−0.29, −0.06] | 0.003 | 2.19 [1.40, 3.43] | 0.001 | 1.26 [0.97, 1.63] | 0.083 |
| | 49-58 | 0.91 [0.85, 0.97] | 0.006 | −0.22 [−0.33, −0.11] | 0.000 | 2.65 [1.72, 4.09] | 0.000 | 1.42 [1.11, 1.82] | 0.006 |
| | 59-68 | 0.85 [0.79, 0.92] | 0.000 | −0.26 [−0.37, −0.14] | 0.000 | 3.19 [2.06, 4.96] | 0.000 | 1.45 [1.12, 1.87] | 0.004 |
| | >68 | 0.78 [0.72, 0.85] | 0.000 | −0.41 [−0.54, −0.29] | 0.000 | 4.40 [2.80, 6.90] | 0.000 | 1.58 [1.20, 2.06] | 0.001 |
| Eye worm episodes (Ref. 0)[a] | 1-5 | 1.00 [0.95, 1.05] | 0.994 | −0.02 [−0.10, 0.06] | 0.601 | 1.03 [0.83, 1.28] | 0.772 | 1.11 [0.95, 1.30] | 0.172 |
| | 6-10 | 0.97 [0.92, 1.02] | 0.262 | −0.04 [−0.12, 0.04] | 0.335 | 1.20 [0.96, 1.50] | 0.117 | 1.11 [0.94, 1.30] | 0.226 |
| | >10 | 0.94 [0.88, 1.00] | 0.061 | −0.11 [−0.21, −0.00] | 0.044 | 1.53 [1.17, 2.00] | 0.002 | 1.21 [0.99, 1.49] | 0.062 |
| | AMD* | 1.00 [0.92, 1.10] | 0.964 | −0.07 [−0.15, 0.01] | 0.077 | 1.15 [0.92, 1.43] | 0.229 | 1.15 [0.88, 1.51] | 0.305 |
| *Trichuris trichiura* infection (Ref. no) | Yes | 0.95 [0.91, 0.99] | 0.013 | | | | | 1.22 [1.06, 1.39] | 0.004 |
| | AMD* | 0.96 [0.89, 1.04] | 0.352 | | | | | 1.10 [0.86, 1.40] | 0.461 |
| Eosinophilia (× 10⁹ cells/L) (Ref. ≤2) | >2 | | | | | | | 0.86 [0.73, 1.01] | 0.064 |
| | AMD* | | | | | | | 1.10 [0.90, 1.34] | 0.356 |
| Main occupation (Ref. Other) | Farmer | 1.05 [1.00, 1.10] | 0.044 | | | | | | |
| Marital status (Ref. as a couple) | Single | 0.92 [0.89, 0.96] | 0.000 | −0.12 [−0.18, −0.06] | 0.000 | 1.16 [0.99, 1.36] | 0.074 | 1.28 [1.14, 1.43] | 0.000 |
| Years of schooling (continuous) | | 1.01 [1.01, 1.02] | 0.000 | 0.02 [0.01, 0.03] | 0.000 | 0.96 [0.94, 0.98] | 0.000 | 0.96 [0.95, 0.98] | 0.000 |

| | | EQautonomy | | EQdaily | | EQpain | | | |
|---|---|---|---|---|---|---|---|---|---|
| | | aIRR [95%CI] | p | aIRR [95%CI] | p | aIRR [95%CI] | p | | |
| Sex (Ref. female) | Male | | | 0.82 [0.71, 0.94] | 0.005 | | | | |
| Age (Ref. 18–28 y.o.) | 29-38 | 7.37 [1.52, 35.70] | 0.013 | 1.86 [1.33, 2.59] | 0.000 | 1.14 [0.92, 1.41] | 0.217 | | |
| | 39-48 | 8.48 [1.80, 39.90] | 0.007 | 1.59 [1.15, 2.21] | 0.005 | 1.23 [1.00, 1.50] | 0.045 | | |
| | 49-58 | 7.46 [1.60, 34.68] | 0.010 | 1.91 [1.40, 2.62] | 0.000 | 1.27 [1.05, 1.54] | 0.015 | | |
| | 59-68 | 8.98 [1.90, 42.51] | 0.006 | 2.14 [1.55, 2.94] | 0.000 | 1.35 [1.11, 1.65] | 0.003 | | |
| | >68 | 10.45 [2.11, 51.69] | 0.004 | 2.66 [1.91, 3.71] | 0.000 | 1.41 [1.14, 1.75] | 0.002 | | |
| Marital status (Ref. as a couple) | Single | 1.74 [1.15, 2.65] | 0.009 | 1.28 [1.13, 1.46] | 0.000 | 1.12 [1.02, 1.22] | 0.020 | | |
| Years of schooling (continuous) | | 0.94 [0.89, 0.99] | 0.025 | 0.96 [0.94, 0.98] | 0.000 | 0.98 [0.97, 0.99] | 0.001 | | |

* AMD: absent/missing data. aIRR: adjusted incidence risk ratio. 95% CI: 95% Confidence interval. MFD: microfilarial density. mf/mL: microfilariae per milliliter of blood. [a] EQvas is an analogic visual scale from 0 (worse disability) to 100 (no disability) and aIRRs < 1 and >1 are thus associated with worse and better scores, respectively. Similarly, EQvts is the "utility index score" ranging from -1 (worst QoL) to +1 (best QoL) and negative and positive ß-coefficients are associated with worse and better scores, respectively. Conversely, for all variables of the five dimensions of the EQ-5D-5L questionnaire, higher scores indicate greater disability and variables with an aIRR > 1 are associated with worse scores. [b] Global Wald test for the eye worm episodes variable: (EQvas, p = 0.331; ESQvts, p = 0.189; EQmobility, p = 0.028; EQanxiety, p = 0.298).

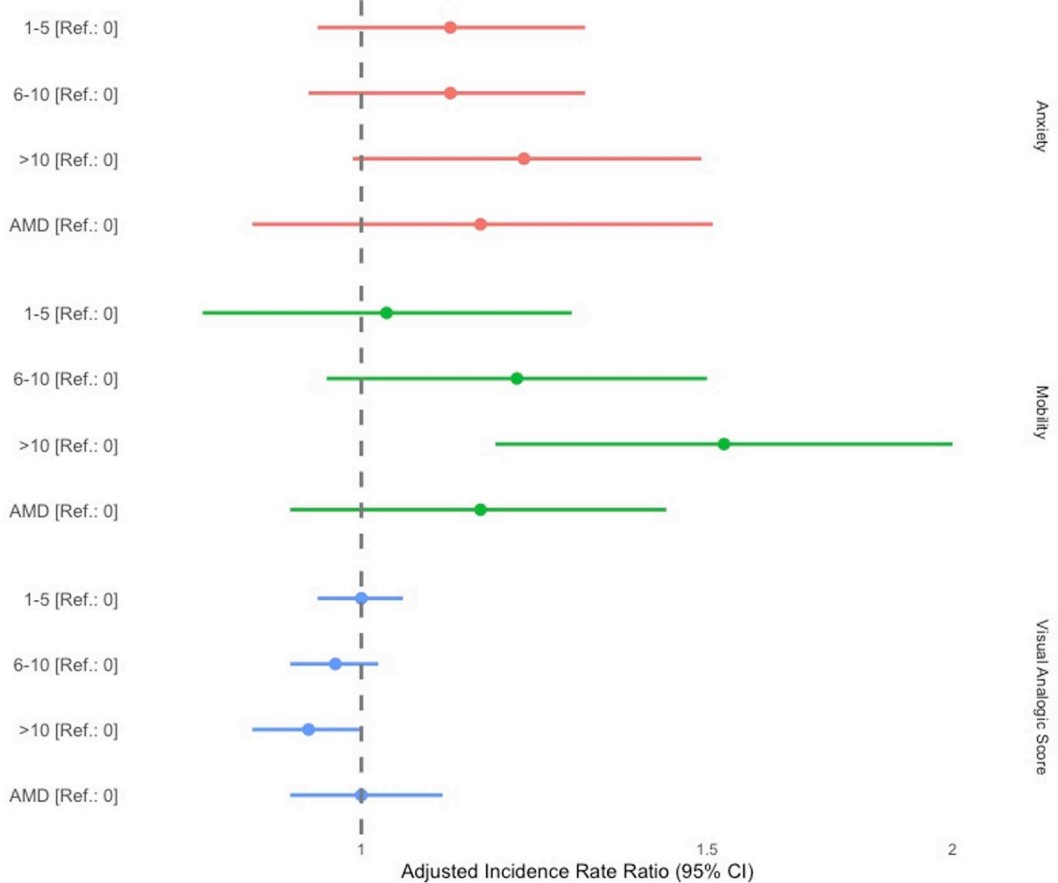

**Fig 4. Associations between eye worm episodes with all our EQ-5D-5L indicators.** The horizontal bars represent 95% confidence intervals, and the dots indicate incidence rate ratios for each category compared to the reference (the reference value for eye worm episodes being 0). These results correspond to Table 3, which shows the final models where the Ew variable was retained ($p < 0.100$).

and QoL in this rural setting. Similarly, sex disparities were evident, with females exhibiting significantly worse scores in several domains compared to males. However, it is noteworthy that this association was not uniform across all indicators, indicating the nuanced nature of sex influences on disability. The higher proportion of disability observed among females compared to males may reflect the overall social organization within these communities. In our study area, females take on the majority of domestic and family tasks, including child-rearing, household maintenance, food provision, and small-scale income-generating activities (subsistence farming and informal trade). In contrast, males tend to focus primarily on village-level responsibilities and agricultural labor, and are significantly less involved in domestic tasks. That said, a potential bias in self-reporting by males cannot be excluded, as they may be less likely to acknowledge limitations in their daily functioning. Dedicated studies exploring gender disparities in disability and daily functioning would be valuable to better understand these dynamics in similar settings. Education level emerged as a protective factor, with more years of schooling consistently associated with better scores across all indicators. This underscores the importance of education as a determinant of health, possibly through its influence on health literacy, access to resources, and socioeconomic status.

In our population, while an overall acceptable average QoL was observed, the analysis of several scores revealed certain disparities. Scores related to the ability to take care of oneself ("WHODAS self-care" and "EQ-autonomy") and to social participation ("WHODAS social") were generally low across the study population, suggesting that these aspects of

functioning were only minimally impacted. Conversely, impaired QoL seems primarily explained by the presence of pain experienced by individuals ("EQpain" variable). The high rank of the "WHODAS household" variable is likely due to a question regarding the ability to perform daily work, which may be influenced by the presence of pain if it exists. Mobility issues also seem to represent a significant burden in this population. Although the weight of the "WHODAS participation" variable appears relatively high, it might be partially explained by the answers recorded for question 5 of the WHODAS questionnaire ("How much have you been emotionally affected by your health problems?" – S1 Text). This question likely reflects a certain level of perceived anxiety, which could be consistent with the high weight observed for the "EQanxiety" variable (Fig 2). The "EQanxiety" result may seem somewhat surprising but one should note that it was only based on a single question of the EQ-5D-5L questionnaire (question 5 of the S2 Text – "I'd like to ask you about anxiety or depression"), while this anxiety issue is not addressed in the WHODAS questionnaire. This item should be interpreted with caution while waiting further psychological studies to better understand this result.

Kato-Katz was the only method used for diagnosing intestinal infections, hampering our ability to detect other potential (protozoan and/or nematode) infections. However, an intriguing result concerns the negative effect of *T. trichiura* infection status on the overall result of the EQ-5D-5L assessment, an effect that seems directly related to the average increase of 22% in the "anxiety" score (aIRR = 1.22). This result warrants further investigation, as suggested earlier, especially since anxiety can also be a symptom of several neuropsychiatric disorders (e.g., depression, psychotic disorders, other somatic conditions). While there are no publications to date clearly reporting a link between *T. trichiura* and anxiety in humans, the impact or alteration of the microbiome and its connection to the brain could provide a possible avenue for exploration. For example, an increase in anxiety has been shown in a murine model of *T. muris* infection [34,35]. Finally, we cannot exclude a possible problem of interpretation on the part of the participants, as infection with *T. trichiura* has sometimes been described as being associated with 'nervousness' [36,37].

Interestingly, despite being a classical biomarker of inflammation and a known contributor to end-organ damage in hypereosinophilic syndromes, eosinophilia was not consistently associated with impaired QoL in our population. This lack of association might reflect the more tolerogenic or compartmentalized nature of eosinophilic responses in chronic parasitic infections, as previously suggested in endemic settings. However, a specific and intriguing result emerged when considering EQanxiety scores: eosinophilia alone was linked to reduced anxiety among individuals with *L. loa* microfilaremia, whereas *T. trichiura* infection alone tended to increase EQanxiety levels. Strikingly, their co-occurrence revealed a statistically significant synergistic effect, with higher anxiety scores than either factor alone. This interaction was not observed in individuals without *L. loa* microfilaremia, though a similar trend was noted. Such findings suggest a complex interplay between immune activation, parasitic burden, and possibly neuroimmune mechanisms influencing mental health outcomes. Further research is warranted to disentangle whether this effect is immunologically mediated, behaviorally perceived, or driven by the microbiome–brain axis.

While some variables, such as a history of Calabar swellings and *L. loa* RDT positivity, did not show statistically significant associations, the frequency of Ew episodes emerged as a substantial predictor of overall disability and mobility impairment, particularly among individuals with a higher number of episodes over their lifetime. Moreover, the average number of days impacted by (mild) disability is increased by 23% in individuals (Table 3, aIRR = 1.23) who had at least 10 Ew episodes during their lifetime, compared to individuals without any Ew episodes during their life, potentially leading to a certain economic impact for the individual. However, history of Ew was not significantly associated with severe QoL impairment (global WHODAS score >50) but "only" with "at least moderate disability" (global WHODAS score >25). Although not significant, but with a p-value <0.100, the presence of a certain gradient effect for the association between the frequency of Ew episodes and EQanxiety may partially be linked to the WHODAS "household" and/or "cognitive" variables. Its association with the "household" variable probably reflects the presence of pain and potential secondary mobility issues. Regarding the "cognitive" variable, we found in a previous study that neurocognitive disorders, assessed using the Montreal Cognitive Assessment (MoCA) score, were significantly associated with *L. loa* MFD but not with history of Ew

episodes [9]. As the MoCA score assesses cognitive impairment more accurately than the tools used during the present study, the results regarding our "cognitive" variable should be interpreted with caution.

The lack of association between between QoL and history of Calabar swellings was somewhat surprising because some episodes can last over several days, be associated with local pain and impair mobility of the wrist. This likely means that (i) spontaneous complications of Calabar swellings such as compression-type peripheral neuropathy [38] are certainly very rare, and/or (ii) pain associated with Calabar swellings does not last long enough and/or is not severe enough to cause significant disability. This lack of association might also be due to classification errors, with participants having difficulties to distinguish Calabar swellings from other conditions. Conversely, the number of past Ew episodes was found to be associated with worse mobility scores, independently of age. As the frequency of Ew migration could be a proxy of the number of adult worms present in the infected persons, one may wonder whether the association between Ew and mobility is due to the episodes themselves or to the number of parasites infecting the host. Regarding the latter, it might be interesting to investigate whether rheumatological manifestations are more frequent in individuals with a high number of past Ew episodes or those with high *L. loa* MFD. Indeed, a number of cases describing either reactive arthritis and/or presence of *L. loa* mf in the synovial fluid have been reported [39,40].

We did not observe any effect of *L. loa* MFD on the individuals' QoL, except in a non-significant manner for the "WHO-DAS social" variable. A survival bias among individuals with the most severe clinical complications associated with high MFD might exist, although this is unlikely. It is more plausible that the QoL in this study population is primarily influenced by pain and mobility issues, and this study suggests the role of adult worms on these scores. However, we may be surprised by the absence of association between *L. loa* MFD and some of the scores which could have been affected if some individuals had chronic disabling conditions such as renal insufficiency or heart disease. Indeed, these conditions could explain the excess mortality associated with *L. loa* MFD [9,10]. This could suggest (i) that the questionnaires we have used are not suitable for evaluating chronic diseases in our study population, or (ii) that individuals with clinical complications associated with *L. loa* MFD remain stable until an acute decompensation occurs. Only future studies will be able to confirm or refute this latter hypothesis.

While studies have statistically demonstrated the negative repercussions of *L. loa* MFD on the functioning of certain human organs [6] and on population mortality rates [10,11], *L. loa* MFD does not seem to have a significant influence on disability in the population. By contrast, *L. loa* adult stages appear to significantly impair the population in their usual activities. Some authors reported that transient painful edema, arthralgia, myalgia, severe headache, transient paralysis of extremities, paresthesia, fatigue, and swellings of any body parts were more significantly associated with loiasis status [5]. However, in contrast to our study, *L. loa* adult worms' history and microfilaremia were gathered to define loiasis status. Based on our results, it is likely that these various symptoms are primarily due to adult worms (as assessed by the Ew question) rather than the direct effect of microfilariae themselves. Apart from fatigue, which remains difficult to interpret, and transient edema, which most likely corresponds to Calabar swellings, we believe that the other reported symptoms could result from the migration of adult worms. In certain anatomical locations, where migration may be restricted due to structural constraints, adult worms could exert pressure on specific neural or vascular structures. For example, if a worm becomes trapped in a confined space, it could press on nerve sheaths or carpal tunnels, potentially leading to paresthesia in the extremities. Similarly, migration in the cervical region could contribute to occipital neuralgia (Arnold's neuralgia), which may explain the observation of severe headaches. Additionally, worms migrating between intermuscular fasciae could be responsible for myalgia. Intra-articular migration, as previously suggested, could lead to synovitis. These mechanisms provide plausible explanations for the variety of symptoms observed in affected individuals. These clinical manifestations cause daily discomfort, leading to meaningful disability in affected individuals. These various results suggest that adult worms may have a clinical impact through peripheral impairments, whereas microfilaremia may have a clinical impact through severe organ involvement, potentially explaining why mortality has been associated only with *L. loa* MFD.

Several limitations to our study should be acknowledged. First, our initial screening procedure in 2019 was based on voluntary participation, which may have introduced a selection bias. However, if such a bias exists, it is likely to affect the overall representativeness of our study population rather than the relationship between our exposure variables and QoL. Second, as this is a cross-sectional study relying on retrospective self-reported data regarding past episodes of Ew and Calabar swelling, recall bias may be present. However, if recall bias exists, it is likely to be non-differential, meaning that it would primarily reduce the statistical power of our results rather than lead to misinterpretation. Importantly, since our participants are part of a longitudinal cohort, we will be able to assess the impact of loiasis on QoL more accurately through follow-up assessments in 2025 and 2028, which will include a reassessment of episode frequency within these intervals, thereby minimizing recall bias. Third, we cannot entirely rule out the possibility of a subtle differential bias whereby individuals experiencing more Ew episodes may associate this symptom with joint pain or daily discomfort, potentially influencing our results. However, we believe such bias to be minimal, as informal discussions with local residents suggest that Calabar swellings should be more commonly associated with several of our indicators, whereas Ew episodes should be primarily linked to reduced visual acuity. Finally, Although TBS is a gold standard technique and was prepared and read by highly experienced technicians in our study, we acknowledge that this method has a lower sensitivity compared to concentration techniques, which could result in the underestimation of microfilaremia prevalence for *L. loa.*

Additionally, our data contained a certain amount of missing data. To address this, we accounted for missing values by including them as a separate category in our models, allowing us to maintain statistical power. However, the observed results for this category may be partly explained by the fact that individuals with missing data were more frequently those who reported living alone—a variable that was itself often significantly associated with worse QoL indicators. While other potential explanatory factors for QoL, such as income level (as a proxy for socioeconomic status), could have been considered, we do not believe that residual confounding is a major concern, or at least, that any existing confounding would have a substantial impact. Nevertheless, the inclusion of such factors could have provided a more comprehensive understanding of QoL within our study population. Lastly, due to the very limited number of studies using WHODAS to assess QoL in the context of NTDs, drawing comparisons with existing literature remains challenging. However, as QoL is strongly influenced by local factors, our findings highlight the need for increased use of standardized tools like WHODAS in Central Africa, including for NTD-related research.

In conclusion, assuming that the frequency of Ew episodes throughout life could serve as a suitable proxy for the number of adult worms present in individuals, the impact of loiasis on daily QoL appears to be primarily attributable to adult worms rather than to MFD. Our findings suggest that adult worms primarily may affect daily activities through peripheral symptoms, such as joint-related discomfort (with notable mobility impairment), while mf primarily induce dysfunction in deep-seated organs. Further research is warranted to better understand the respective clinical impacts of adult and microfilarial stages of *L. loa.*

## Supporting information

**S1 Text. WHODAS 2.0 questionnaire.**
(DOCX)

**S2 Text. EQ-5D-5L questionnaire.**
(DOCX)

**S1 Table. Disabilities for the 6 domains from the WHODAS 2.0 questionnaire.**
(DOCX)

**S2 Table. Descriptive results for WHODAS 2.0 questionnaire for the number of days impaired during the last month.**
(DOCX)

**S3 Table. Saturated multivariable analyses on global WHODAS 2.0 score and the related six domains.**
(DOCX)

**S4 Table. Disabilities for the 6 dimensions from the EQ-5D-5L questionnaire.**
(DOCX)

**S5 Table. Saturated multivariable analyses on EQ-5D-5L scores (EQmobility, EQautonomy, EQdaily, and EQpain).**
(DOCX)

**S6 Table. Saturated multivariable analyses on EQ-5D-5L scores (EQvas, EQvts, and EQanxiety).**
(DOCX)

**S1 Fig. Relationship between scores and number of years of schooling.**
(DOCX)

**S2 Fig. Forest plot of the final models for WHODAS 2.0 questionnaire.**
(DOCX)

**S3 Fig. Forest plot of the final models for EQ-5D-5L questionnaire.**
(DOCX)

## Acknowledgments

We thank the French Embassy in Republic of Congo. We thank the Lékoumou health district, the medical, paramedical and technical staff of the Sibiti hospital, the *Programme National de Lutte contre l'Onchocercose* (PNLO) and IRD drivers, and the participants for agreeing to participate.

## Author contributions

**Conceptualization:** Sébastien D. S. Pion, Michel Boussinesq, Cédric B. Chesnais.

**Data curation:** Jérémy T. Campillo, Valentin Dupasquier, Elodie Lebredonchel, Ludovic G. Rancé, Glorifié Madoulou Moulabou, Sébastien D. S. Pion, Michel Boussinesq, François Missamou, Cédric B. Chesnais.

**Formal analysis:** Samuel Beneteau, Cédric B. Chesnais.

**Funding acquisition:** Cédric B. Chesnais.

**Investigation:** Jérémy T. Campillo, Elodie Lebredonchel, Sébastien D. S. Pion, Michel Boussinesq, François Missamou, Cédric B. Chesnais.

**Methodology:** Jérémy T. Campillo, Sébastien D. S. Pion, Michel Boussinesq, Cédric B. Chesnais.

**Project administration:** François Missamou, Cédric B. Chesnais.

**Supervision:** Jérémy T. Campillo, Ange Clauvel Niama, Richard R. Bileckot, François Missamou, Cédric B. Chesnais.

**Validation:** Cédric B. Chesnais.

**Visualization:** Cédric B. Chesnais.

**Writing – original draft:** Marlhand C. Hemilembolo.

**Writing – review & editing:** Jérémy T. Campillo, Valentin Dupasquier, Elodie Lebredonchel, Samuel Beneteau, Ange Clauvel Niama, Sébastien D. S. Pion, Michel Boussinesq, Cédric B. Chesnais.

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
