## [Decision Letter · Decision Letter 0]

29 Jan 2025

PNTD-D-24-01363

Disability and quality of life assessment using WHODAS-12 items 2.0 and EQ-5D-5L in a rural area endemic for loiasis in the Republic of Congo: a population-based cross-sectional study (the MorLo project)

Dear Dr. Chesnais,

Thank you for submitting your manuscript to PLOS Neglected Tropical Diseases. After careful consideration, we feel that it has merit but does not fully meet PLOS Neglected Tropical Diseases's publication criteria as it currently stands. Therefore, we invite you to submit a revised version of the manuscript that addresses the points raised during the review process.

Please submit your revised manuscript within 60 days Mar 30 2025 11:59PM. If you will need more time than this to complete your revisions, please reply to this message or contact the journal office at plosntds@plos.org. Please include the following items when submitting your revised manuscript:

We look forward to receiving your revised manuscript.

Kind regards,

Aysegul Taylan Ozkan, M.D., Ph.D.,

Academic Editor

Guilherme Werneck

Section Editor

Shaden Kamhawi

co-Editor-in-Chief

Paul Brindley

co-Editor-in-Chief

**Journal Requirements:**

At this stage, the following Authors/Authors require contributions: Hemilembolo C Marlhand, Campillo T Jérémy, Dupasquier Valentin, Lebredonchel Elodie, Beneteau Samuel, Rancé G Ludovic, Madoulou Moulabou Glorifié, Niama Ange, Bileckot R Richard, Pion DS Sébastien, Boussinesq Michel, Cédric B. Chesnais, and Missamou François. Please ensure that the full contributions of each author are acknowledged in the "Add/Edit/Remove Authors" section of our submission form.

- TM on page: 7.

3) We notice that your supplementary Tables, and information are included in the manuscript file. Please remove them from the main file. Supporting files should be uploaded separately with the file type 'Supporting Information'. Please ensure that each Supporting Information file has a legend listed in the manuscript after the references list.

4) We note that your Data Availability Statement is currently as follows: "Anonymized data will be hosted on the https://dataverse.ird.fr/ server, and its terms of use will be those in force on the hosting site." Please confirm at this time whether or not your submission contains all raw data required to replicate the results of your study. Authors must share the “minimal data set” for their submission. PLOS defines the minimal data set to consist of the data required to replicate all study findings reported in the article, as well as related metadata and methods (https://journals.plos.org/plosone/s/data-availability#loc-minimal-data-set-definition).

**Comments to the Authors:**

**Please note that one of the reviews is uploaded as an attachment.**

**Reviewers' Comments:**

Reviewer's Responses to Questions

**Key Review Criteria Required for Acceptance?**

**Methods**

-Are the objectives of the study clearly articulated with a clear testable hypothesis stated?

-Is the study design appropriate to address the stated objectives?

-Is the population clearly described and appropriate for the hypothesis being tested?

-Is the sample size sufficient to ensure adequate power to address the hypothesis being tested?

-Were correct statistical analysis used to support conclusions?

-Are there concerns about ethical or regulatory requirements being met?

Reviewer #1: The study objectives—to assess quality of life and disability in a loiasis-endemic region—are clearly stated. However, the hypothesis regarding the different impact of adult worms compared to microfilarial density on QoL could have been described more explicitly in the manuscript.

The cross-sectional design is appropriate to the objectives, although it has inherent limitations for inferring causality. These limitations are correctly acknowledged by the authors.

The target population is well-defined, with proper inclusion and exclusion criteria. Stratification of participants by microfilaremic status and symptom history aligns well with the research objectives. However, providing more details about how participants were recruited and addressing potential selection biases would improve the methodology section.

The sample size of 991 participants seems sufficient. While power calculations and effect size estimations for the key outcomes (WHODAS and EQ-5D-5L scores) are mentioned, they could have been further elaborated to increase transparency.

The use of Poisson and Negative Binomial regression models is appropriate given the nature of the data. The manual backward selection process is valid, but the authors should explain better how they accounted for potential confounders and interactions.

The handling of missing data (over 20% in some variables) requires a more explicit discussion to understand the impact on the findings.

It would also be interesting to include details about the training of interviewers and the translation or adaptation process for the questionnaires.

Reviewer #2: -Are the objectives of the study clearly articulated with a clear testable hypothesis stated? YES

-Is the study design appropriate to address the stated objectives? NO

-Is the population clearly described and appropriate for the hypothesis being tested? NO

-Is the sample size sufficient to ensure adequate power to address the hypothesis being tested? YES

-Were correct statistical analysis used to support conclusions?

-Are there concerns about ethical or regulatory requirements being met YES

**Results**

-Does the analysis presented match the analysis plan?

-Are the results clearly and completely presented?

-Are the figures (Tables, Images) of sufficient quality for clarity?

Reviewer #1: The results are clearly presented, although some tables appear overly detailed. Consolidating information or combining variables into summary tables would enhance clarity for readers.

Figures and tables are of good quality, but simplified labels and improved readability would further strengthen the presentation. For example, the term “MD” is used for both “missing data” and “mild disability,” which can confuse readers depending on the table or section.

Statistical significance could also be visually highlighted in a more consistent manner across the figures.

Reviewer #2: Confoundin factors such as STH carriage, subjective symptoms of loaisis should be better presented.

the participants should be classified according to the type of parasitic infection as well as monoparasitism or multiple parasite carriage

**Conclusions**

-Are the conclusions supported by the data presented?

-Are the limitations of analysis clearly described?

-Do the authors discuss how these data can be helpful to advance our understanding of the topic under study?

-Is public health relevance addressed?

Reviewer #1: The conclusions are well-supported by the presented data. The novel finding that adult worms significantly impact QoL more than microfilarial density is important and contributes meaningfully to the field.

The study’s limitations, including its reliance on self-reported symptoms and the cross-sectional design, are appropriately acknowledged. However, more discussion on how these limitations might influence the interpretation of results would be beneficial.

The manuscript addresses an important gap in the understanding of QoL and disability in loiasis-endemic regions, highlighting the public health burden of the disease. The discussion connects the findings effectively to practical implications for patient care and policy-making.

Reviewer #2: -Are the conclusions supported by the data presented? not entirely

-Are the limitations of analysis clearly described? NOT SUFFICIENTLY

-Do the authors discuss how these data can be helpful to advance our understanding of the topic under study? YES

-Is public health relevance addressed? NO

**Editorial and Data Presentation Modifications?**

Reviewer #1: The manuscript is generally well-organized, but the following improvements are suggested to improve clarity and presentation:

Improve the visual presentation of the results.

Add subheadings in the results section for better organization.

Optimize table formats to improve readability.

Include a participant flow diagram to help readers understand how participants were selected and analysis process.

Reviewer #2: MAJOR REVISION (SEE THE ATTACHED FILE)

**Summary and General Comments**

Reviewer #1: The use of standardized QoL tools such as WHODAS 2.0 and EQ-5D-5L is a significant strength, providing validated metrics for cross-study comparisons.

The large sample size improves the generalizability of the findings within the loiasis-endemic region.

However, recall bias may have influenced the self-reported outcomes, such as Calabar swellings and worm passage history. Additionally, the absence of a discussion on socioeconomic factors, which could confound QoL and disability results, could be a relevant limitation.

This manuscript represents a valuable addition to the field, offering insights into the QoL impact of loiasis and bridging a knowledge gap in NTD and disability research.

It is suggested to expand the discussion of potential mechanisms, provide additional context on prior QoL studies in NTDs, further elaborate on the implications for control programs, and strengthen the discussion on the unexpected T. trichiura findings.

Reviewer #2: SEE

PLOS authors have the option to publish the peer review history of their article (what does this mean? ). If published, this will include your full peer review and any attached files.

**Do you want your identity to be public for this peer review?** For information about this choice, including consent withdrawal, please see our Privacy Policy .

Reviewer #1: No

Reviewer #2: No

**Figure resubmission:**
---

## [Decision Letter · Decision Letter 1]

18 May 2025

PNTD-D-24-01363R1

Disability and quality of life assessment using WHODAS-12 items 2.0 and EQ-5D-5L in a rural area endemic for loiasis in the Republic of Congo: a population-based cross-sectional study (the MorLo project)

Dear Dr. Chesnais,

Thank you for submitting your manuscript to PLOS Neglected Tropical Diseases. After careful consideration, we feel that it has merit but does not fully meet PLOS Neglected Tropical Diseases's publication criteria as it currently stands. Therefore, we invite you to submit a revised version of the manuscript that addresses the points raised during the review process.

Please submit your revised manuscript within 60 days Jun 17 2025 11:59PM. If you will need more time than this to complete your revisions, please reply to this message or contact the journal office at plosntds@plos.org. Please include the following items when submitting your revised manuscript:

We look forward to receiving your revised manuscript.

Kind regards,

Aysegul Taylan Ozkan, M.D., Ph.D.,

Academic Editor

Guilherme Werneck

Section Editor

Shaden Kamhawi

co-Editor-in-Chief

Paul Brindley

co-Editor-in-Chief

**Journal Requirements:**

1) We have noticed that you have uploaded Supporting Information files, but you have not included a complete list of legends. Please add a full list of legends for your Supporting Information files after the references list. Please ensure that the labels of the supplementary files match their citations and legends. For example S4 table is labeled as Appendix S4 and S10 Figure is labeled as Appendix S10.

2) Regarding Figure 1, thank you for stating "Adapted from 7." Please include the source details in the figure legend.

3) We note that you have indicated that there are restrictions to data sharing for this study. PLOS only allows data to be available upon request if there are legal or ethical restrictions on sharing data publicly. For more information on unacceptable data access restrictions, please see https://journals.plos.org/plosntds/s/data-availability#loc-unacceptable-data-access-restrictions.

b) If there are no restrictions, please upload the minimal anonymized data set necessary to replicate your study findings to a stable, public repository and provide us with the relevant URLs, DOIs, or accession numbers. For a list of recommended repositories, please see https://journals.plos.org/plosone/s/recommended-repositories. You also have the option of uploading the data as Supporting Information files, but we would recommend depositing data directly to a data repository if possible.

**Comments to the Authors: **

**Please note that one of the reviews is uploaded as an attachment.**

**Reviewers' Comments:**

Reviewer's Responses to Questions

**Key Review Criteria Required for Acceptance?**

**Methods**

-Are the objectives of the study clearly articulated with a clear testable hypothesis stated?

-Is the study design appropriate to address the stated objectives?

-Is the population clearly described and appropriate for the hypothesis being tested?

-Is the sample size sufficient to ensure adequate power to address the hypothesis being tested?

-Were correct statistical analysis used to support conclusions?

-Are there concerns about ethical or regulatory requirements being met?

Reviewer #1: (No Response)

Reviewer #3: NA.

Reviewer #4: The objectives and methods are clearly stated. The population is well described and appropriate to the hypothesis. The samples size is adequate. No concerns about ethical requirements

Reviewer #5: The Methods have been reviewed by previous referees and I have no further comments on these.

**Results**

-Does the analysis presented match the analysis plan?

-Are the results clearly and completely presented?

-Are the figures (Tables, Images) of sufficient quality for clarity?

Reviewer #1: (No Response)

Reviewer #3: NA.

Reviewer #4: These appear to be post-hoc analyses. The tables are overly detailed and difficult to follow. Some have text far too small to be legible. There is an over-reliance on acronyms in the text of the manuscript that makes it very difficult to follow. For example, FD, PD, MB, MBP, HbAS, aIRR, etc. It would be clearer if the actual terms were used, especially where there may be confusion.

The figure legends are insufficient. In particular, figure 2 is unclear--do black and grey represent male and female or NO history of eye worm vs >10 episodes of eye worm? If the latter, the intermediate group (1-9 episodes of eye worm) should also be shown. Captions for figures 3 and 4 are missing (as are captions for the figures in the appendix).

Reviewer #5: The Figures need some attention and I have described the issues in the attached file.

**Conclusions**

-Are the conclusions supported by the data presented?

-Are the limitations of analysis clearly described?

-Do the authors discuss how these data can be helpful to advance our understanding of the topic under study?

-Is public health relevance addressed?

Reviewer #1: (No Response)

Reviewer #3: NA.

Reviewer #4: Many of the conclusions are based on conjecture rather than data. For example, the primary message of the paper is that poorer disability or quality of life scores are associated with having >10 eye worm episodes. Poorer scores are also highly correlated with increasing age. Yet there is no discussion of the relationship between having >10 eye worm episodes and increasing age. Obviously, the longer one lives and is infected, the more eye worm episodes one will experience. Does the correlation with eye worm episodes simply reflect the age of those with >10 episodes?

Reviewer #5: Yes.

**Editorial and Data Presentation Modifications?**

Reviewer #1: (No Response)

Reviewer #3: NA.

Reviewer #4: 1. n the abstract and introduction, the word "would", which connotes a hypothetical or counterfactual state, is used in situations not intended to be counterfactual (i.e. "would be infected", "would be a frequent reason", "would mainly affect daily activities", etc. Consider revision.

2. Figure 1 "126 excluded (<500 microfilaremia per mL) is nonsensical. Microfilaremia is a condition, not a parasite stage.

3. line 349 "none significant interaction"

Reviewer #5: I have made a number of editorial suggestions listed in the attached document.

**Summary and General Comments**

Reviewer #1: I appreciate the authors' effort in addressing the recommendations from the initial review. The revisions have significantly improved the clarity and methodological rigor of the manuscript, particularly in explicitly stating the hypothesis, providing more details on participant selection, and handling missing data. The inclusion of new figures and a flowchart has enhanced the readability of the results, while the expanded discussion section has provided a more comprehensive analysis of the findings, especially regarding the differential impact of adult worms and microfilarial density on quality of life. Given these improvements, I consider the manuscript suitable for publication.

Reviewer #3: NA.

Reviewer #4: This manuscript is a useful attempt to further define the effect of loiasis on disability and quality of life among those infected. It employs resonable methods and the statistical analyses seem sound. However, there are some aspects that make it difficult to follow, and the conclusions are based on conjecture rather than data.

Specific comments:

1. These data do not show that adult worm burden causes the quality of life issues. They show a correlation with >10 eye worm episodes, which could be age-related as discussed above.

2. Some methods need further explanation:

line 149: how were discrepancies in Mf count defined? Were the slides read twice by one technician or once each by two technicians?

Explanatory variables (line 240): Why was age a categorical variable instead of a continuous one? If there were 10 age categories used, why are only 6 shown in the tables and figures? Why were eye worm episodes categorical rather than continuous?

3. There seem to be 51 or more degrees of freedom in the multivariate model (including 10 age categories). Is this appropriate? Why not conserve degrees of freedom by analyzing the appropriate input variables as continuous rather than categorical?

4. Why convert the WHO-DAS-2 scores to a scale of 100 rather than using the actual, validated score? Was this done simply by multiplying by a conversion factor? If so, why was this necessary? If there is a good rationale and the scale of 100 scores are retained, they should at least be referred to as "adjusted" or "modified" WHO-DAS-2 scores.

5. line 407 "we estimated" suggests that the numbers presented are a guess. Why not simply report that the model-adjusted scores WERE as reported?

6. Do Table 2 and Table 3 show ALL the variables included in the multivariable analysis?

7. Could the association with reported episodes of eye worm simply represent a tendency on the part of some individuals to over-report symptoms?

8. Line 483. What basis is there for the claim that WHO-DAS-2 is particularly suited for rheumatologic conditions? If there is support for this claim, it should be cited.

9. Line 500: "It appears that factors related to....were only marginally affected." Affected by what? Is there a comparison here, or are you just saying that scores in these areas across the whole study population were normal?

10. The discussion theorizes that adult worms are causing mobility impairment, but if >10 eye worm episodes are found only in older individuals, this may have nothing to do with worms and may only represent age. An analysis of the correlation between eye worm episodes and age would be helpful. Line 553: "the number of past Ew episodes was found to impact mobility" is incorrect, since it assumes causation, which is not proven. Ew episodes were only correlated with mobility. Same issue line 580. Finally, the conclusion that "adult worms primarily affect daily activities through peripheral symptoms, such as joint-related discomfort (with notable mobility impairment)" is not warranted. Again, this study shows association, not causation, and this statement should be revised.

Reviewer #5: This is an important paper that had already been reviewed by two competent referees. I read the revised manuscript and the replies to referees by authors. I found that the authors addressed carefully and point-by-point the referees' comments and queries. The Figures and Supporting Information need attention. I also suggest that 3-4 references be added. I have made editorial suggestions in an effort to improve the clarity and conciseness in places, as well as as the grammar where necessary.

PLOS authors have the option to publish the peer review history of their article (what does this mean? ). If published, this will include your full peer review and any attached files.

**Do you want your identity to be public for this peer review?** For information about this choice, including consent withdrawal, please see our Privacy Policy .

Reviewer #1: No

Reviewer #3: No

Reviewer #4: No

Reviewer #5: No

**Figure resubmission:**
---

## [Decision Letter · Decision Letter 2]

3 Jul 2025

PNTD-D-24-01363R2Disability and quality of life assessment using WHODAS-12 items 2.0 and EQ-5D-5L in a rural area endemic for loiasis in the Republic of Congo: a population-based cross-sectional study (the MorLo project)PLOS Neglected Tropical Diseases Dear Dr. Chesnais, Thank you for submitting your manuscript to PLOS Neglected Tropical Diseases. After careful consideration, we feel that it has merit but does not fully meet PLOS Neglected Tropical Diseases's publication criteria as it currently stands. Therefore, we invite you to submit a revised version of the manuscript that addresses the points raised during the review process. Please submit your revised manuscript within 30 days Aug 02 2025 11:59PM. If you will need more time than this to complete your revisions, please reply to this message or contact the journal office at plosntds@plos.org.  Please include the following items when submitting your revised manuscript:* A rebuttal letter that responds to each point raised by the editor and reviewer(s). You should upload this letter as a separate file labeled 'Response to Reviewers '. This file does not need to include responses to any formatting updates and technical items listed in the 'Journal Requirements' section below.* A marked-up copy of your manuscript that highlights changes made to the original version. You should upload this as a separate file labeled 'Revised Manuscript with Track Changes '.* An unmarked version of your revised paper without tracked changes. You should upload this as a separate file labeled 'Manuscript '. If you would like to make changes to your financial disclosure, competing interests statement, or data availability statement, please make these updates within the submission form at the time of resubmission. Guidelines for resubmitting your figure files are available below the reviewer comments at the end of this letter. We look forward to receiving your revised manuscript. Kind regards, Aysegul Taylan Ozkan, M.D., Ph.D.,Academic EditorPLOS Neglected Tropical Diseases Guilherme WerneckSection EditorPLOS Neglected Tropical Diseases

Shaden Kamhawi

co-Editor-in-Chief

Paul Brindley

co-Editor-in-Chief

 **Journal Requirements:**  1) Please provide a completed 'Competing Interests' statement, including any COIs declared by your co-authors. If you have no competing interests to declare, please state "The authors have declared that no competing interests exist". 

 **Reviewers' comments:**  Reviewer's Responses to Questions

**Key Review Criteria Required for Acceptance?**

**Methods**

-Are the objectives of the study clearly articulated with a clear testable hypothesis stated?

-Is the study design appropriate to address the stated objectives?

-Is the population clearly described and appropriate for the hypothesis being tested?

-Is the sample size sufficient to ensure adequate power to address the hypothesis being tested?

-Were correct statistical analysis used to support conclusions?

-Are there concerns about ethical or regulatory requirements being met?

Reviewer #5: Yes.

Reviewer #6: (No Response)

**Results**

-Does the analysis presented match the analysis plan?

-Are the results clearly and completely presented?

-Are the figures (Tables, Images) of sufficient quality for clarity?

Reviewer #5: Yes.

Reviewer #6: Tables and figures comments:

1. Table 1:

-would be clearer if you had vertical line separating data columns for WHODAS vs EQvts vs EQvas

-there should be further discussion of statistical differences between the data shown ex. Differences in the different age groups for each type of score

2. Most figures need further explanation in the legends.

3. Fig 2: explain light vs dark gray color for the two bars (assume one is male and one female)

-Explain what is on the Y axis (% disability) and how it was calculated. I could not find this in the methods. I at first assumed this was the % of the summed global score from each domain, but the global score was also only 20-30%.

4. Table 2: why no aIRR for Sex and cognitive score?

5. Fig S7 is supposed to have the raw scores for each EQ dimension on scale of 1-5. However, listed are scores that range from 4-49? What is the mean listed in S7? This is unclear to me

**Conclusions**

-Are the conclusions supported by the data presented?

-Are the limitations of analysis clearly described?

-Do the authors discuss how these data can be helpful to advance our understanding of the topic under study?

-Is public health relevance addressed?

Reviewer #5: Yes.

Reviewer #6: (No Response)

**Editorial and Data Presentation Modifications?**

Reviewer #5: The authors have addressed all my previous comments. My recommendation is 'Accept'.

Reviewer #6: Clarity and grammatical recommendations

-Line 48: more typical to say associated with, rather than associated to

-Line 52-53: awkward sentence (notable mobility impairment contributing to mobility impairment)

-line 99-100: might be clearer as loiasis (defined as either microfilaremia or a history of adult worms, based on eye worm or Calabar swelling)

-Line 189: Awkward: recommend substituting “data points” for “informations”

-lines 214-218: would clarify here that you are measuring days with each level of disability (not a certain score indicating a level of disability as done elsewhere. Might be more clear if written: “They were asked to provide the number of days the difficulties were present (termed H1, indicating mild disability)…

-line 387-392: awkwardly written. Could change to The scores recorded for WHODAS 2.0 selfcare, WHODAS 2.0 social, and EQautonomy showed that these domains and dimension represented a very low proportion of disability.

Whereas other parameters had a higher percent of disability (rather than a preoccupation)

-line 402: would clarify: number of partial disability days increased with the frequency of Calabar swellings

-Lines 471 and 473: would change to worse anxiety (or mobility) scores rather than impact on anxiety.

General Comments

1. It might be useful to specify the language or languages that informed consent was obtained in (line 125)

2. Line 190: Some background on Raploa might be useful for those outside the Loa field

3. Line 366-373: should cite which figure presents this continuous variable data, or if the raw data is not presented in the paper, possibly another supplemental figure with a scatter plot showing score vs year of schooling could be added (with each different category in a different color)

4.The text headings separate out WHODAS 2 and then later EQ-5D-5L for descriptive and multivariate analyses, but the text combines description of both WHODAS 2 and EQ-5D-5L.

5. Additional description of the raw data in table 1 would be useful. There is basically no description of the table 1 results, only the continuous results which are not presented in a figure or table.

6. Table 1 could benefit from statistics – there are only statistics for the multivariate analysis not the raw data in table 1. For example, it is interesting and useful to show no statistical difference in scores based on eosinophilia.

7. Paper could benefit from discussion of how Mf might change over time and skew results. Mf count was not repeated during 2022 enrollment and was based on 2019 assessment.

On further reflection, it is unclear if the TBS was done off blood from 2022 or the original 2019 assessment and which is listed in table 1.

8. Abstract conclusions focus most on mobility as does the concluding paragraphs, but Fig 2 demonstrates high scores for mobility, but also pain and anxiety and daily living. This should be further explained. I think it is also a little unclear what they mean by peripheral symptoms in the abstract

9. The main conclusion in the abstract talks about impact of adult worms (rather than EW episodes) on mobility. This has been raised by a previous reviewer. I think it should be stated more clearly that the main correlation between worse QOL scores is with number of eye worm episodes, which they are assuming to be a proxy for number of adult eye worms. It also only highlights mobility, even though their data seems to indicate issues with mobility, pain, anxiety and other concerns as well.

10. The paper overall is dense and difficult to read given the large number of different scores they are referencing and their very different scales. It is very helpful for there to be reminders of what each scale represents with each table, such that someone could read just the table and legend and understand the data presented. This would be redundant but I think overall helpful to the flow of the manuscript. The full description follows table 2 and 3 but not table 1 (should add WHODAS description to table 1)

11. I appreciate the good discussion of the surprising result that Mf count does not appear correlated with worse QOL, despite being correlated with organ disfunction and death. There is much written about the chronic eosinophilia in loiasis. I think there should be more discussion of the lack of correlation between eosinophilia and QOL, as well as how the cutoff of 2 was chosen.

**Summary and General Comments**

Reviewer #5: The authors have addressed all my previous comments. My recommendation is 'Accept'.

Reviewer #6: The paper addresses an important clinical question: the impact of loiasis on overall wellbeing. This infection impacts millions of people and has been substantially neglected in terms of research into clinical course and treatment discovery. Further, several recent studies have demonstrated clear associations between loiasis and end organ damage and increased mortality. Therefore, this study looking at how loiasis impacts overall quality of life is clinically interesting and important. They also delve into which aspects of this parasitic infection are most correlated with wellness (ex microfilaria or signs more associated with adult worms). However, the paper is overall difficult to read given the many different scoring systems used. Have made some suggestions to help improve clarity and presentation.

PLOS authors have the option to publish the peer review history of their article (what does this mean? ). If published, this will include your full peer review and any attached files.

**Do you want your identity to be public for this peer review?** For information about this choice, including consent withdrawal, please see our Privacy Policy .

Reviewer #5: No

Reviewer #6: No

---

## [Editor Report · Decision Letter 3]

20 Aug 2025

Dear Dr Chesnais,

We are pleased to inform you that your manuscript 'Disability and quality of life assessment using WHODAS-12 items 2.0 and EQ-5D-5L in a rural area endemic for loiasis in the Republic of Congo: a population-based cross-sectional study (the MorLo project)' has been provisionally accepted for publication in PLOS Neglected Tropical Diseases.

Best regards,

Aysegul Taylan Ozkan, M.D., Ph.D.,

Academic Editor

Guilherme Werneck

Section Editor

Shaden Kamhawi

co-Editor-in-Chief

Paul Brindley

co-Editor-in-Chief

---

## [Editor Report · Acceptance letter]

Dear Dr Chesnais,

We are delighted to inform you that your manuscript, "Disability and quality of life assessment using WHODAS-12 items 2.0 and EQ-5D-5L in a rural area endemic for loiasis in the Republic of Congo: a population-based cross-sectional study (the MorLo project)," has been formally accepted for publication in PLOS Neglected Tropical Diseases.

Best regards,

Shaden Kamhawi

co-Editor-in-Chief

Paul Brindley

co-Editor-in-Chief
